# Investigating the assimilation of CALIPSO global aerosol vertical observations using Four-Dimensional Ensemble Kalman Filter

Yueming Cheng[1,2], Tie Dai[1,2*], Daisuke Goto[3], Nick A. J. Schutgens[4], Guangyu Shi[1,2], and Teruyuki Nakajima[5]

[1]Collaborative Innovation Center on Forecast and Evaluation of Meteorological Disasters, Nanjing University of Information Science and Technology, Nanjing, China
[2]State Key Laboratory of Numerical Modeling for Atmospheric Sciences and Geophysical Fluid Dynamics, Institute of Atmospheric Physics, Chinese Academy of Sciences, Beijing, China
[3]National Institute for Environmental Studies, Tsukuba, Japan
[4]Faculty of Science, Free University of Amsterdam, Amsterdam, Netherlands
[5]Earth Observation Research Center, Japan Aerospace Exploration Agency, Tsukuba, Japan

*Corresponding author:* Tie Dai (daitie@mail.iap.ac.cn)

**Abstract.** The aerosol vertical information is critical to quantify the influences of the aerosol on the climate and environment, however large uncertainties still persist in model simulations. In this study, the vertical aerosol extinction coefficients from the Cloud-Aerosol Lidar with Orthogonal Polarization (CALIOP) onboard the Cloud–Aerosol Lidar and Infrared Pathfinder Satellite Observation (CALIPSO) are assimilated to optimize the hourly aerosol fields of the Non-hydrostatic ICosahedral Atmospheric Model (NICAM) online coupled with the Spectral Radiation Transport Model for Aerosol Species (SPRINTARS) using the four-dimensional Local Ensemble Transform Kalman Filter (4D-LETKF). A parallel assimilation experiment using the bias-corrected Aerosol Optical Thicknesses (AOTs) from the Moderate Resolution Imaging Spectroradiometer (MODIS) is conducted to investigate the effects of assimilating the observations whether including the vertical information on the model performances. Additionally, an experiment simultaneously assimilating both the CALIOP and MODIS observations is conducted. The assimilation experiments are successfully performed for a one-month long, making it possible to evaluate the results in a statistical sense. The hourly analyses are validated via both the CALIOP observed aerosol vertical extinction coefficients and the AOT observations from the MODIS and AErosol RObotic NETwork (AERONET). Our results reveal both the CALIOP and MODIS assimilations can improve the model simulations. The CALIOP assimilation is superior to the MODIS assimilation in modifying the incorrect aerosol vertical distributions and reproducing the real magnitudes and variations, and the joint CALIOP and MODIS assimilation can further improve the simulated aerosol vertical distribution. However, the MODIS assimilation can better reproduce the AOT distributions than the CALIOP assimilation and the inclusion of the CALIOP observations has an insignificant impact on the AOT analysis. This is probably due to the nadir-viewing CALIOP has much sparser coverages than the MODIS. The assimilation efficiencies of CALIOP decrease with the increasing distances of the overpass time, indicating that more aerosol vertical observation platforms are required to fill the sensor-specific observation gaps and hence improve the aerosol vertical data assimilation.

## 1 Introduction

Aerosols have significant impacts on the air quality, climate change, radiation balance, and hydrological cycle (Charlson et al., 1992; Huang et al., 2014; Liu et al., 2011, 2014; Nakajima et al., 2001; Ramanathan et al., 2001). Aerosols may also contribute to the regional differences in the historical warming rates (Huang et al., 2017). Due to the large uncertainties in the parameterizations of the various aerosol processes such as emission, transport, and deposition, it is still a challenge for model itself to accurately quantify the effects of aerosols on the earth system (Huneeus et al., 2011; Liu et al., 2015; Jia et al., 2015; Myhre et al., 2013; Sato et al., 2018; Sato and Suzuki, 2019; Textor et al., 2006). Aerosol data assimilation, which makes optimal use of both observations and numerical simulations to obtain the best possible estimates of aerosol behaviors, is an emerging way for getting accurate predictions and characterizations of atmospheric aerosol and its influence.

Recently, aerosol assimilation studies have been conducted to improve the model simulation performances generally using one of the optimal interpolation, variational, and ensemble-based methods. The assimilated aerosol observations commonly include the space-borne or ground-based aerosol optical thicknesses (AOTs) from the POLarization and Directionality of the Earth's Reflectances (POLDER) (Generoso et al., 2007), Moderate Resolution Imaging Spectroradiometer (MODIS) (Dai et al., 2014a; Di Tomaso et al., 2017; Hyer et al., 2011; Liu et al., 2011; Yin et al., 2016a, 2016b; Yumimoto et al., 2011; Yumimoto et al., 2017; Zhang and Reid, 2006), Himawari-8 (Dai et al., 2019; Yumimoto et al., 2018), AERONET (Schutgens et al., 2010a, 2010b), and the multi-sensor (Rubin et al., 2017; Zhang et al., 2014). Although assimilating AOTs can strongly constrain the horizontal distributions of the aerosol burdens, there are limited in improving the vertical information of aerosols. The vertical distribution of aerosols is a combined characteristic of atmospheric transport patterns, residence times in the atmosphere and the efficiency of the vertical energy exchange (Koffi et al., 2012; Yu et al., 2010), which shows the large diversities up to an order of magnitude among models (Textor et al., 2006). Reductions of the large uncertainties in the aerosol vertical distributions are of crucial importance for the accurate evaluation of the aerosol impacts on the earth system such as the aerosol direct effect (Oikawa et al., 2013, 2018) and the aerosol indirect effect (Liu et al., 2019a, 2019b).

Apart from the AOTs, the aerosol vertical information such as the aerosol extinction coefficient is another variable for aerosol assimilation. Uno et al. (2008) and Yumimoto et al. (2008) developed a four-dimensional variational (4D-Var) assimilation system with a regional dust model to assimilate the retrieved-extinction coefficient from the ground-based National Institute for Environment Studies (NIES) Lidar network for investigating Asian dust. Due to the limited spatio-temporal frequencies of the aerosol vertical observations and the noise signal from the lidar observations, the aerosol vertical assimilation is still at an initial stage. The Cloud–Aerosol Lidar and Infrared Pathfinder Satellite Observation (CALIPSO), which carried the Cloud-Aerosol Lidar with Orthogonal Polarization (CALIOP) instrument with high horizontal and vertical resolutions, exhibits great potential for reducing the uncertainties in the global spatio-temporal distributions of aerosols especially the vertical profiles. The CALIPSO satellite provides an outstanding opportunity for studying aerosol vertical information especially over the high elevation and harsh climate regions (Huang et al., 2007). Based on

the CALIOP observations, a new technique was developed to distinguish anthropogenic dust from natural dust for exploring the effects of anthropogenic emission on radiative forcing, climate change, and air quality (Huang et al., 2015). A coupled 2D/3D-Var system which assimilated the CALIOP-retrieved aerosol extinction profiles for two weeks was developed by Zhang et al. (2011). Zhang et al. (2014) also conducted the multi-sensor experiments and found that the inclusion of CALIOP data can improve the aerosol vertical distribution and hence forecasted advection. The variational data assimilation systems use the pre-calculated static model error covariance matrix, whereas the ensemble-based Kalman filter (EnKF) can generate a flow-dependent model error covariance matrix to better represent the background error (Evensen, 1994). The flow-dependent model errors are calculated by the ensemble simulations. Sekiyama et al. (2010) initially applied the four-dimensional Local Ensemble Transform Kalman filter (4D-LETKF) to assimilate the CALIOP attenuated backscattering coefficients with the analysis at the center of the 48 h assimilation window. However, the hourly aerosol analyses using the 4D-LETKF approach can provide more accurate aerosol features for investigating the environmental and climate effects of aerosols (Dai et al., 2019).

Therefore, in the present study, we apply the 4D-LETKF and an aerosol model named Spectral Radiation Transport Model for Aerosol Species (SPRINTARS) online coupled with a new dynamical atmospheric model called Non-hydrostatic ICosahedral Atmosphere Model (NICAM) to generate hourly aerosol horizontal and vertical analyses for one-month using the CALIOP aerosol extinctions. The results are validated using both the CALIOP extinctions and the MODIS and AERONET AOTs observations. To the best of our knowledge, this is probably the first study to conduct the hourly aerosol vertical extinctions assimilation using four-dimensional ensemble Kalman method for one month.

The observation data used in this study are described in Sect. 2. The forward model and the data assimilation methodology are described in Sect. 3. Section 4 presents the assimilation results and their validations. The discussions and the main conclusions achieved in this study are shown in Sect. 5 and Sect. 6, respectively.

## 2 Observational Data

### 2.1 CALIOP

CALIOP, a space-borne two-wavelength polarization lidar that is the prime payload instrument carried by the CALIPSO satellite, probes the high-resolution vertical structures and properties of clouds and aerosols since 2006 (Winker et al., 2007). CALIPSO flies as one of 5 satellites in the so-called "A-train" constellation of satellites, and all the satellites are in the 705 km sun-synchronous polar orbit with a 16-day repeat cycle. During both day and night, the CALIOP continuously provides vertical profiles of aerosol extinction coefficient at 532 nm and 1064 nm, with a uniform spatial resolution of 60 m vertically and 5 km horizontally over an altitude range of -0.5 km to 30 km. In this study, we only use the aerosol extinction coefficients at 532 nm in the CALIPSO lidar level 2 (L2) Version 4.10 aerosol profile products over the altitude range below 10 km for assimilation (http://www-calipso.larc.nasa.gov/). The CALIPSO lidar level 2 Version 4.10 Vertical Feature Mask (VFM)

products which consist of information on feature type and subtype are used for aerosol discrimination. The AOTs from the CALIPSO level 3 Version 3 aerosol profile monthly products at 2°×5° resolution for cloud-free conditions in November from 2006 to 2016 are also used.

## 2.2 MODIS

The MODIS aboard both NASA's Terra and Aqua satellites is making near-global daily observations of the earth in a wide spectral range (0.41-15 μm) (Remer et al., 2005). In this study, the United States Naval Research Laboratory (NRL) quality-assured and controlled MODIS level 3 AOT products are also used for data assimilation. The NRL MODIS AOTs are based on the MODIS level 2 aerosol products and aggregated to 1°×1° grid with reduced biases and error variances for using in the near-

real-time data assimilation processes (Zhang and Reid, 2006). The datasets consist of 6-hourly gridded AOTs and error estimates at four times per day (00:00, 06:00, 12:00, and 18:00 UTC). Since the representation of the observations are considered during the development of the NRL MODIS datasets and the NRL MODIS AOTs have been subjected to extensive quality assurance and quality check procedures for aerosol assimilation (Zhang and Reid, 2006; Zhang et al., 2008), we directly use the

NRL MODIS AOTs and the corresponding AOT uncertainties every 6-hour at 1° without further operation. The MODIS-Aqua Collection 6.1 level 2 Dark Target (DT)/Deep Blue (DB) merged AOT observations at 550 nm are used to validate the model simulations. The DT and DB merged AOT provides a more gap-filled dataset than the observation which is only retrieved from the individual algorithms (https://modis.gsfc.nasa.gov) (Levy et al., 2013; Sayer et al., 2014). The Community

Intercomparison Suite (CIS) tool (Watson-Parris et al., 2016), which was developed to allow the straightforward comparison of remote sensing, in situ and model data, is used to aggregate the level 2 MODIS AOTs at 10 km and 5 minutes resolution to produce the hourly MODIS AOTs at 2°×2°for comparison. The AOTs from the MODIS-Aqua Collection 6.1 (C6.1) level 3 monthly aerosol products with a 1°×1° resolution in November from 2006 to 2016 are also used.

**2.3 AERONET**

The AErosol RObotic NETwork (AERONET; http://aeronet.gsfc.nasa.gov/) Version 3 (V3) data provides globally distributed near-real-time observations of aerosol spectral optical thickness at nominal standard aerosol wavelengths as 340, 380, 440, 500, 675, 870, 1020, and 1640 nm (Holben et al., 1998; Giles et al., 2019). Due to the both availability of the AOTs at 440 nm in most AERONET

sites and the model calculations, we directly use the level 2 cloud-screened AERONET AOTs at 440 nm for the validation. The AERONET retrieved instantaneous AOTs within the preceding 1h are averaged to calculate the hourly mean AOTs for comparison.

## 3 Model and Data Assimilation Methodology

### 3.1 Model

A global three-dimensional aerosol transport model SPRINTARS (Takemura et al., 2000, 2003, 2009) online coupled with NICAM (Satoh et al., 2005, 2008, 2014; Tomita and Satoh., 2004) is used as the

forward model to predict the aerosol spatial and temporal evolutions. The major tropospheric aerosols including dust, sea salt, sulfate, and carbonaceous aerosols are simulated in NICAM-SPRINTARS considering the main aerosol processes including emissions, advection, dry and wet depositions. A three-dimensional icosahedral grid advection scheme preserving monotonicity and consistency with continuity for aerosol transport is adopted in NICAM (Niwa et al., 2011). The dust and sea salt aerosols spanning wide size ranges are divided into 10 and 4 bins for transport, respectively, whereas the sulfate and carbonaceous aerosols are predicted using the bulk modal method (Takemura et al., 2000). The aerosol-cloud-radiation interactions are included in the aerosol-coupled version of NICAM (Sato et al., 2018). In this study, the NICAM-SPRINTARS is set up with a homogeneous horizontal resolution about 223 km and a vertical resolution of 40 layers from surface to approximately 40 km altitude. The vertical grid spaces approximately 160 m near the surface, and the spaces exponentially increase to approximately 1320 m around 16 km. NICAM is successfully applied to produce a high-resolution global simulation with a horizontal resolution of about 0.87 km (Miyamoto et al., 2013), indicating a potential application of a sub-kilometer global aerosol simulation and data assimilation when the computer resources are available. With the stretched-grid system, NICAM can also be used to run by partially high resolution simulations in the object regions (Goto et al., 2015). Therefore, there is a future prospect to extend the present assimilation method to fine scale regional analyses using the stretched-grid system implemented in NICAM (Uchida et al., 2016, 2017). The biomass burning and anthropogenic emission inventories for BC, OC, and $SO_2$ used in this study are from the Global Fire Emissions Database (GFEDv3.1; van der Werf et al., 2010) and the Hemispheric Transport of Air Pollution (HTAP; Janssens-Maenhout et al., 2015) as same as in Dai et al. (2019). The dust and sea salt emission fluxes are both mainly depended on the near-surface wind speeds (Dai et al., 2018; Takemura et al., 2009). The meteorological fields in NICAM are nudged by the reanalysis data every 6 h from the NCEP Final (FNL) Analysis (https://rda.ucar.edu/datasets/ds083.2/) to reduce the influences of the uncertainties in the meteorological conditions on the aerosol simulations.

**3.2 Data Assimilation Methodology**

The 4D-LETKF (Dai et al., 2019; Hunt et al., 2007) assimilation system of the NICAM_SRPINTARS is used to assimilate the CALIOP vertical extinctions and NRL MODIS AOTs observations, which is an extension of the LETKF to assimilate the asynchronous observations. In this section, we briefly describe the basic scheme of LETKF and more details on the LETKF algorithm and its implementation can be found in Hunt et al. (2007). The LETKF determines the analysis ensemble mean $\bar{x}^a$ locally in the space spanned by the ensemble as the following formula

$$\bar{x}^a = \bar{x}^b + X^b \bar{w}^a \tag{1}$$

where $\bar{x}^b$ and $X^b$ represent background (forecast) ensemble mean and background ensemble perturbations, respectively. The weight matrix $\bar{w}^a$ is obtained as

$$\bar{w}^a = \tilde{P}^a (Y^b)^T R^{-1} (y^o - \bar{y}^b) \tag{2}$$

where the matrix $R$ is the observation error covariance matrix; the $y^o$ and $\bar{y}^b$ represent the assimilated observations and the ensemble mean background observations; the ensemble background observations are calculated by applying the observation operator $H$ to the ensemble member state vector $x^{b(i)}$ as

$y^{b(i)} = H(x^{b(i)})$; and the matrix $Y^b$ represents ensemble background observation perturbations, whose $i$th column is $y^{b(i)} - \bar{y}^b, \{i = 1, 2, ..., k\}$ with $k$ ensemble members. The analysis error covariance in ensemble space is calculated as

$$\tilde{P}^a = \left[(k-1)I - Y^{b^T}R^{-1}Y^b\right]^{-1} \tag{3}$$

where $I$ is the identity matrix and the analysis ensemble perturbations $X^a$ are obtained as

$$X^a = X^b[(k-1)\tilde{P}^a]^{1/2} = X^bW^a \tag{4}$$

whose $i$th column is $x^a(i) - \bar{x}^a, \{i = 1, 2, ..., k\}$. The analysis ensemble by adding $\bar{x}^a$ to each of the columns of $X^a$ forms the optimal initial conditions for the ensemble forecast to produce the background for the next analysis.

To generate the hourly aerosol analysis, LETKF requires hourly switching between the model ensemble forward simulations and assimilation of the synchronous observations. However, as shown in Fig. 1, 4D-LETKF considers approximate model trajectories by linear combinations of the background ensemble trajectories and compare these approximate trajectories with the observations taken over the assimilation time window (Hunt et al., 2007), which can assimilate asynchronous observations and

avoid frequent switching between assimilations and model ensemble forecasts. With respect to the assimilation window time of 24 hours, the system performs the ensemble forecast for 24 hours and outputs at every hour time slot within the time window. Based on the innovations throughout the assimilation window, the ensemble mean background observations and the background ensemble perturbation matrix are formed at the various time slots when the observations are available and then

vertically concatenated to form a combined background observation mean $\bar{y}^b$ and perturbation matrix $Y^b$. Following to the LETKF formulas, all the innovations $(y^o - \bar{y}^b)$ and $Y^b$ throughout the day are used for the calculation of the weight matrix $\bar{w}^a$ and $W^a$. The weights determined at the end of a short assimilation window (e.g., 24 hours) should be valid throughout the window (Dai et al., 2019; Hunt et al., 2007; Kalnay and Yang, 2010). To perform a linear combination of ensemble trajectories, the same

$\bar{w}^a$ is then applied to the state vector at every hour time slot throughout the assimilation window to obtain the hourly analysis ensemble mean (Di Tomaso et al., 2017; Dai et al., 2019). The analyzed aerosol fields at the last slot can then directly serve as the initial conditions for the next 24 hours of forward simulation, therefore, our implementation of the 4D-LETKF can also avoid a multiple use of one observation without overlapping the ensemble forecasts between adjacent assimilation cycles. One

advantage of the 4D-LETKF is the localization technique, which allows the local analyses to choose different linear combinations of the ensemble members in local regions using a prescribed localization scale to explore a much higher-dimensional space than the ensemble space and reduce the spurious correlations with distance (Hunt et al., 2007). Furthermore, localization enables parallel computation to reduce computational cost (Yumimoto et al., 2018). The horizontal and vertical localization are both

performed in the observational error covariance matrix by the physical distance to avoid discontinuity in the analysis (Miyoshi et al., 2007). The observational error covariance matrix is calculated as the observation uncertainties multiple the inverse of the horizontal and vertical localization factors, that the effect of observation on analysis is descending by the distance increasing. The horizontal and vertical localization factors are both defined following Gaussian shapes as $exp(-r^2/2\sigma^2)$, where the $\sigma$

represents the localization length and the $r$ is the distance of observations from the local patch center. Although the Gaussian function has infinitely long tails, we truncate the tails to simulate the fifth-order piecewise rational function by Gaspari and Cohn (1999), which is widely used localization weighting function in the EnKF studies (Miyoshi et al., 2007). The fifth-order rational function drops to zero at

$r = 2 \cdot \sqrt{10/3} \cdot \sigma$ and we do not assimilation observations beyond this distance. We do not apply the temporal localization in this study since our results indicate the temporally remote asynchronous observations within 24 hours have limited influence on the analysis. To better reduce the computational resources and limit the model information to a few modes only, the subspecies of the SPRINTARS predicted aerosols are summarized into a fine (carbonaceous and sulfate aerosol) and a coarse (sea salt

and dust aerosol) mode for assimilation as same as in Schutgens et al. (2010a). The modeled aerosol fine and coarse mass mixing ratios are hourly optimized using the hourly aggregated observations during the assimilation window and the mass mixing ratios after data assimilation for the each subspecies are determined from their relative fractions before assimilation (Dai et al., 2019). The observation operators we used to map the model state vector into the aerosol extinction coefficient $\sigma$

and the AOT observation space at wavelength $\lambda$ are calculated as $\sigma_j(\lambda) = \beta(\lambda) \cdot M_j$ and $AOT(\lambda) = \sum_{j=1}^{n}\big(\beta(\lambda) \cdot M_j\big)$, where the $M_j$ represents the simulated aerosol dry mass concentration in each model level $j$, $\{j = 1,2,\ldots,n\}$ and the $\beta$ represents the prescribed aerosol mass extinction coefficient (Dai et al., 2014b).

A total of 668 CALIPSO orbit paths in November 2016 are obtained and used for the aerosol data

assimilation. To identify the aerosol signals and screen out the cloud signals, we applied several quality control procedures to remove the noisy or highly uncertain observations before aggregating the profile data. The quality controls include (1) the vertical feature mask must be determined as aerosols; (2) the Cloud-Aerosol Discrimination (CAD) score must be lower than -80, indicating high confidence of discriminating as aerosols; (3) Extinction Quality Control (QC) flag must be equal to 0 or 1 (Young

and Vaughan, 2009); (4) the extinction coefficient must be greater than 0 and less than 100 km $^{-1}$; (5) the uncertainty of extinction coefficient must be lower than 10 km $^{-1}$, indicating the stable iteration.

After the quality control of the data, we aggregate the original CALIOP extinction coefficients to the model grid boxes. Firstly, we perform the hourly horizontal aggregation to the model horizontal resolution about 2°×2° in the CALIOP observations level (i.e., every 0.06 km) using the mean value of

all the reasonable sub-grid observations within $\pm30$ minutes. The CALIOP extinction observations within the range of $\mu \pm \sigma$ are used for the average, where $\mu$ and $\sigma$ represent the mean and the standard deviation of the sub-grid observations in the aggregation cell respectively. To avoid assimilation of sub-grid features likely to create anomalous feature in the horizontal 2° grid cell (Schutgens et al., 2017), the number of the used retrievals within 2° grid cell must be also greater than 20 and the

coefficient of extinction variations within the grid cell (i.e., standard deviation/mean) must be less than 0.5 as similar as Zhang and Reid. (2006). Then, the horizontally regridded observations within each model layer are averaged to serve as the assimilated observations. The observation uncertainties are assumed to be 20% of the mean aerosol extinctions in reference to Winker et al. (2007) and Sekiyama et al. (2010).

As summarized in Table 1, a total of five numerical experiments are conducted for this study. Data assimilation is initiated at 00:00 UTC on 1 November 2016 and terminated at 00:00 UTC on 1 December 2016. The initial condition at 00:00 UTC on 1 November is prepared by a one-month simulation which is executed by NICAM-SPRINTARS without any aerosol data assimilation as a spin-

up. A single deterministic simulation with the default model configuration is performed as a reference in November 2016 (a free running, FR experiment hereafter). The two assimilation experiments assimilating the CALIOP vertical extinction coefficients with time windows of 1 and 24 hr (hereafter called the LETKF-CALIPSO and 4D-LETKF-24H-CALIPSO, respectively) are performed to investigate the influences of the assimilation time window on the hourly aerosol analysis. Combined

with the 4D-LETKF-24H-CALIPSO experiment, two more assimilation experiments are designed to investigate the impacts of assimilating the observations whether including the vertical information and the impacts of the multi-sensor data assimilation on the model simulations. The first one assimilates the NRL MODIS AOTs including no vertical aerosol information (4D-LETKF-24H-NRL experiment hereafter). In the second one, the CALIOP vertical extinction coefficients and NRL MODIS AOTs are

assimilated simultaneously. The assimilation system parameters are based on several tuning experiments as discussed in Sect. 5. Twenty members with spatiotemporally perturbing aerosol emissions are used to generate the model ensemble simulations. The perturbation factors follow lognormal distributions as same as those used in Dai et al. (2019), except the uncertainty of sea salt is assumed as 500% following Yumimoto and Takemura (2011). This is due to the emission fluxes of sea

salt are still not well known, which can span a factor about 20 for the different source functions even excluding outliers (Grythe et al., 2014). The horizontal and vertical localization lengths are set as 200 km and one model layer (increasing by the altitude), respectively. To prevent filter divergence, the multiplicative inflation with a fixed factor of 1.1 is performed on the background ensemble following Sekiyama et al. (2010).

**4 Results**

The results in the FR, LETKF-CALIPSO, 4D-LETKF-24H-CALIPSO, and 4D-LETKF-24H-NRL experiments are firstly compared with the assimilated CALIOP vertical extinctions and the NRL MODIS AOTs as the self-verification respectively. In order to investigate the similarities and the differences among the four assimilation experiments, we use three independent observations (i.e.,

AERONET AOTs, independent MODIS Aqua AOTs, and independent day-time CALIOP aerosol extinctions) to validate the simulated AOTs and aerosol extinctions, respectively. To quantify the model performances, the statistical criteria (Boylan and Russell, 2006; Willmott et al., 2012; Yumimoto et al., 2017), including the mean fractional bias (MFB), the mean fractional error (MFE), the root mean square error (RMSE), the correlation coefficient (CORR) and the index of agreement

(IOA), are calculated between the simulated results and observations. Because of the small differences between the AOTs at 532 nm and 550 nm, i.e., between 2% and 4% for typical Angstrom exponents of 0.5 to 1 (Kittaka et al., 2011), we directly use the modeled aerosol extinction coefficients at 550 nm to compared with the observations at 532 nm.

## 4.1 Internal Check of the Analysis

### 4.1.1 Internal Check of the Analysis With the Assimilated CALIOP Aerosol Extinctions

In this section, we only analyse the performance of the experiments assimilating CALIOP extinctions using the assimilated CALIOP extinction coefficients at 532 nm as sanity or internal checks (Benedetti et al., 2009). Figures 2a-j show the scatter plots of the assimilated CALIOP extinction coefficients versus the simulated ones over the global land and ocean, and Figures 2k-l further show the probability distribution functions (PDFs) of collocated forecast-minus-observation deviations in the FR experiment and analysis-minus-observation deviations in the assimilation experiments. Based on the model performance evaluation statistical metrics (i.e., MFB, MFE, RMSE, CORR, and IOA), the two CALIOP assimilation experiments are evidently better than the FR experiment especially over the ocean regions. The CORR values are 0.317 and 0.365 over the global land and ocean for the FR experiment, whereas the LETKF-CALIPSO and 4D-LETKF-24H-CALIPSO experiments significantly improve the CORRs over global land (ocean) to 0.600 (0.792) and 0.668 (0.782) respectively. The distributions of the extinction coefficient deviations in the FR experiment show the systematically negative biases, indicating the model tends to underestimate the extinction coefficients in most of the world. Obviously, the PDFs over both the land and ocean in the two CALIOP assimilation experiments are symmetrical to the value of 0 and more squeezed with higher peaking. Merely 13.56% (27.49%) and 15.73% (32.03%) of the extinction coefficient deviations over the land and ocean are within $\pm 0.01 (\pm 0.02)$ in the FR experiment, while 37.89% (56.30%) and 51.15% (73.91%) of the deviations are achieved within $\pm 0.01 (\pm 0.02)$ in the LETKF-CALIPSO experiment. The performances of the 4D-LETKF-24H-CALIPSO experiment are generally comparable to those of the LETKF-CALIPSO experiment. This indicates the weights determined at the end of the 24 hours assimilation window are valid to optimize the ensemble trajectories and the temporally remote asynchronous observations within 24 hours have limited influence on the analysis. The eliminations of the strong underestimations over both the global land and ocean in the two CALIOP assimilation experiments provide a positive sanity check of the assimilation system.

To further assess the effects of assimilating CALIOP aerosol extinctions on the aerosol column simulations over different aerosol regimes, we choose thirteen regions and classify them into four groups according to their surface aerosol emissions regimes (Huang et al., 2013). The column integrated aerosol extinctions from 0 to 10 km altitude is used for comparison. The four groups are dominated by the biomass burning smoke, dust, industrial pollution, and marine aerosols, respectively (Fig. 3a). As shown in Figs. 3b-f, the modeled column integrated extinctions are compared with the CALIOP ones over the thirteen selected regions. All the five statistical metrics of the LETKF-CALIPSO and 4D-LETKF-24H-CALIPSO experiments are superior to those of the FR experiment, indicating that the spatio-temporal distributions of the column integrated aerosol characteristics are closer to those of the CALIOP observations after the CALIOP assimilation. The modeled column integrated extinctions in the FR experiment show the negative biases compared to CALIOP for 11 out of the 13 regions, reaching up to 42% and 66% for the SEA and SAM regions in the biomass burning regime, respectively. With the CALIOP assimilation, the modeled column integrated extinctions are generally increased over all the underestimated regions. The RMSE value decreases from 0.028 to

0.020 (0.022) and the CORR value increases from 0.871 in FR experiment to 0.955 (0.963) in the LETKF-CALIPSO (4D-LETKF-24H-CALIPSO) experiment. In addition to the evaluation of the modeled mean column integrated extinctions in Figs. 3b-f, we further employ the Taylor graph (Taylor, 2001) to quantitatively assess the pattern correspondence between the simulated and observed fields on the regional level (Fig. 3g). The correlation coefficient, the centered pattern root mean square (RMS) difference, and the ratios of the modeled and observed standard deviation are all indicated by a single point on a two-dimensional plot. The ratios of the modeled and observed standard deviation in the FR experiment range from 0.30 over the SAM region to 2.05 over the ECN region, while the upper bounds are significantly down to 1.59 and 1.19 in the LETKF-CALIPSO and 4D-LETKF-24H-CALIPSO experiments. The regions with obviously underestimated dispersion are less improved by the aerosol data assimilation as shown by the relatively small variations of the lower bounds. This is due to the aerosol extinctions are generally strongly underestimated over the regions with underestimated dispersion, making it difficult to generate enough model errors by perturbing aerosol emissions. Thus, the observational errors are too large relative to the model errors, which translates to a reduced impact of the observation on the model state.

Figures 4-6 show the scatter plots of the modeled and the CALIOP observed hourly extinction coefficients within 0-10 km over the thirteen regions for the FR, LETKF-CALIPSO, and 4D-LETKF-24H-CALIPSO experiments, respectively. The aerosol extinctions are classified into 4 altitude ranges (0-0.5 km, 0.5-1.0 km, 1.0-2.0 km, and 2.0-10.0 km) to further investigate the effects of the CALIOP assimilation on the simulations of the aerosol vertical characteristics. The statistical metrics of the LETKF-CALIPSO and 4D-LETKF-24H-CALIPSO experiments are clearly superior to those of the FR experiment over all the thirteen regions, and those of the 4D-LETKF-24H-CALIPSO experiment are generally comparable to those of the LETKF-CALIPSO experiment. The slight differences between the two CALIOP assimilation experiments are induced by the effects of the asynchronous observations on the hourly analysis. The large negative MFB values reveal the FR experiment tends to underestimate the aerosol extinction coefficients over all the three biomass burning regions especially over the SAM region. The improvements of the CALIOP assimilation on the simulated extinction coefficients over the SAM region are not so obvious compared to the other two regions. This is probably due to the emission fluxes of biomass burning aerosols over the SAM region are underestimated obviously, leading to an underestimation of the model uncertainty and, hence, the analysis underweights the observations. The extinction coefficients in the free atmosphere (2-10 km) of the LETKF-CALIPSO and 4D-LETKF-24H-CALIPSO experiments are more narrowly distributed along the 1:1 line than those in the boundary layer (0-2 km) for the SAF and SEA regions. The lower observed extinction coefficients associated with the lower observation errors in the higher altitude and the high injections of fire product induce the model uncertainties are relatively larger than the observation ones, hence, the analysis underweights the model. Dust is a predominant component of aerosol in the NAF, WCN, and WAU regions, where the main deserts in the world are located in. The improvements of the simulated extinction coefficients by the CALIOP assimilation in the NAF and WAU regions are more obvious than those in the WCN region. The proportions of observations within 1-2 km (2-10 km) for the NAF, WCN, and WAU regions are 37% (11%), 22% (22%), and 24% (14%), respectively. This indicates that

the dust aerosols emitted from the desert in West China have the higher possibilities in transporting in the relatively high altitude than those emitted from the desert in North Africa (Ginoux et al., 2001; Huang et al., 2009). Therefore, the dust emission in West China will have wider impacts on the downwind areas than that of NAF, leading to a smaller model spread over the dust source regions of

WCN and, hence, the analysis underweights the observations. In the WAU region, the significant overestimations of the extinction coefficients over the 1 km in the FR experiment are probably caused by a simulated dust storm (Dai et al. 2019), and the CALIOP assimilation corrects this overestimation. In the IND, ECN, WEU, and EUS regions, urban and industrial aerosols are the major part of the aerosol loadings (Penning de Vries et al., 2015). The aerosols in this regime especially over ECN

mainly exist below 2 km as indicated by the relatively large extinction coefficients. It is apparent that the CALIOP assimilation can significantly improve the model performances of simulated extinction coefficients over all the four anthropogenic aerosol regions. The NWP, NAT, and CAT regions are oceanic regions located downwind of the major dust and industrial pollution sources. Thus, these oceanic regions are substantially influenced by the mixture of ocean emissions, the ship exhaust, and

the transported continental emissions (Sorooshian and Duong, 2010). It is apparent the CALIOP assimilation over the maritime downwind regime has the highest assimilation efficiencies among the four regimes. A common problem that the analyses generally fail to correct the significantly underestimated extinction coefficients is found over all the four aerosol regimes. This is probably due to the insufficient emissions lead to the underestimated extinctions and model errors, thus the analysis

underweights the observation.

The vertical profiles of the monthly and regional averaged aerosol extinctions and the ratios between the modeled and observed standard deviations of the aerosol extinctions and the collocated CALIOP retrievals over the thirteen regions are shown in Fig. 7. We ignore the levels where the number of the available CALIPSO observations is less than 10. With respect to the biomass burning regime, the FR

experiment underestimates the aerosol extinction profiles over all the available levels of all the three biomass burning regions. This indicates the biomass burning emissions in November 2016 are probably stronger than those used in this study. The shape of the simulated extinction profile over the SEA region in the FR experiment is generally consistent to the CALIOP observations, whereas the FR experiment fails to simulate the descending trend of the extinction coefficients over the SAF region.

The CALIOP assimilation experiments decrease the negative biases over all the three biomass burning regions and achieve the descending trend of the extinction coefficients over the SAF region. Moreover, the ratios of the simulated and observed standard deviation over the SAF region with the CALIOP assimilation are closer to 1 than those in the FR experiment. With respect to the dust regime, it is found the CALIOP assimilation has marginal impacts on the vertical profiles except over the WAU region.

The simulated profile of the extinction coefficients in the over the WAU is apparently more comparable to the CALIOP observed one than that in the FR experiment, and the significantly descending trend in the CALIOP observations below 1 km are successfully reproduced in the experiments with CALIOP assimilation. The simulated ascending trend of the aerosol extinctions above 2.5 km over the WAU in the FR experiment is corrected by the CALIOP assimilation, which is

probably due to the elimination of the simulated dust event. There are limited improvements with data

assimilation over the WCN region due to the limited observations to be assimilated. For the NAF region, the CALIOP assimilation induces a slightly larger negative bias of the vertical profile than that of the FR experiment. This is due to the assimilation is more efficient to reduce the positive biases than the negative ones. This situation is also found in the IND region. Over the ECN region, it is obvious

that the FR experiment significantly overestimates the extinction coefficients below 1 km. This is due to the anthropogenic emission inventories used in this study are based on the year 2010, whereas the anthropogenic emissions over ECN have been significantly reduced due to the national regulations of the anthropogenic emissions. The CALIOP assimilation successfully eliminates the significant overestimations below 1 km. Over the WEU region, the CALIOP assimilation experiments reproduce

the vertical profile of the aerosol extinctions much better than those in the FR experiment. Over the EUS region, the overestimations of the extinction coefficients over the 1 km (mainly contributed by the sulfate aerosols) are correctly amended by the data assimilation. Over NWP and NAT regions, although the FR experiment simulates totally opposite vertical distributions of extinction coefficients to those of the CALIOP observations, the CALIOP assimilation experiments reproduce the profile of the

CALIOP observations successfully. The significant discrepancies of the aerosol profiles over the maritime downwind regions between the FR and the CALIOP assimilation experiments are due to the relatively sufficient spread of the sea salt emissions below 2 km (figure not shown for brevity).

**4.1.2 Internal Check of the Analysis With the Assimilated NRL MODIS AOTs**

In this section, we perform the self-verification of the simulated hourly AOTs in the 4D-LETKF-24H-

NRL experiment over the whole month through a comparison with the assimilated NRL MODIS AOTs. Figures 8a-j show the spatial distributions of the biases and RMSEs between the simulated and the NRL MODIS AOTs at 550 nm. In Fig. 8k, we also present the PDF plots of the AOT deviations between the simulated hourly AOTs and the NRL MODIS observed ones over the whole month. The 4D-LETKF-24H-NRL experiment significantly reduces the positive biases and the associated high

RMSEs over southeast China and Australia. As shown in Fig. S1, the improvements over southeast China mostly benefit from the reductions of sulfate, OC and BC aerosols, and the improvements over Australia mostly benefit from the reductions of natural dust aerosols. The significant underestimation of the simulated AOTs in FR experiment over Northern India is corrected by the increments of the anthropogenic aerosols after the MODIS assimilation. The distribution of the AOT deviations relative

to the NRL MODIS observations for the FR experiment is negatively biased and the 4D-LETKF-24H-NRL experiment is superior to the FR experiment as indicated by the reduced biases and the peaks in 0. The frequencies of AOTs deviation within $\pm 0.05$ ($\pm 0.10$) in the 4D-LETKF-24H-NRL experiment are 71.83% (90.32%), whereas those in the FR experiment are 64.73% (83.29%) .

**4.2 Independent Validation of the Analysis With AERONET AOTs**

As shown in Figs. 9a-j, we further present the maps of the statistical metrics (biases and RMSEs) between the modeled and the AERONET observed AOTs at 440 nm calculated over the whole month at each AERONET stations as the independent validation for the five experiments. We select an AERONET site if it has simultaneously at least 10 hours in one month (not necessarily consecutive)

where the hourly AOTs is not missing. In order to make the statistics significant, we require at least 10 observations in each selected site. A total of 191 AERONET sites are selected for comparison. Due to the sparse CALIOP observations and the localization used in the assimilation system, the simulated AOTs in many model grids are unable to be affected by the aerosol data assimilation. Therefore, we only analyse the simulated AOTs with more than 30% variations between the 4D-LETKF-24H-CALIPSO and FR experiments hereafter. The bias in the FR experiment indicates that the model tends to underestimate the AOTs over 60% stations in the world but overestimate the AOTs over the polluted industrial regions such as Eastern America, Western Europe, South and East Asia. The strongest biases and highest RMSEs in the FR experiment are both found in South and East Asia, and this is due to large uncertainties in the temporal and spatial distributions of the anthropogenic aerosol emissions. The four assimilation experiments especially the experiments including MODIS observations clearly reduce the biases and RMSEs over the regions such as Eastern America and Western Europe. The 4D-LETKF-24H-NRL experiment decreases the bias and RMSE over 140 (73%) and 153 (80%) of the total available 191 AERONET sites, respectively. The LETKF-CALIPSO and 4D-LETKF-24H-CALIPSO experiment decrease the biases in 58% and 63% sites, respectively. This indicates that the CALIOP assimilation using 4D-LETKF method is slightly superior to the one using LETKF method. We also give the PDF plots of the AOT deviations between the simulated hourly AOTs and the AERONET observed ones for the five experiments. The negatively biased PDF for the FR experiment also reveals that the simulated AOTs are underestimated globally. The biases are improved as the peaks nearer to 0 with both the CALIOP and MODIS assimilations, and the performance of the two assimilation experiments including MODIS observations are better than the two experiments only with CALIOP assimilation. The frequency of AOT deviations within ±0.05 (±0.10) increases from 46.25% (65.96%) in the FR experiment to 57.89% (75.90%) in the 4D-LETKF-24H-NRL experiment and 57.35% (76.38%) in the 4D-LETKF-24H-CALIPSO-NRL experiment. This indicates that the inclusion of the CALIOP data has an insignificant impact on the AOT analysis, which is also mentioned in Zhang et al. (2014). This is because the MODIS observations include more aerosol column information than the CALIOP observations.

Figure 10a shows the detailed comparisons of the temporal evolutions of the hourly AOTs at 440 nm for the five experiments with the AERONET-retrieved ones over the site named Pushan_NU. Compared to the simulated AOTs in the FR experiment, the modeled AOTs in the assimilation experiments are much closer to the AERONET observed ones, especially reducing the significant overestimations of AOTs from 3 to 7 November. The overestimations in the FR experiment are mainly induced by the sulfate productions due to aqueous phase conversion from $SO_2$. The CALIOP derived vertical aerosol sub-types (Fig. 10m) are dominated by the sea salt and other aerosols, proving the simulated sulfate productions in the FR experiment are incorrect. In fact, there are no CALIOP orbit paths pass the Pushan_NU site, indicating the improvements of the AOTs over 3 to 7 November can benefit from the assimilation of the CALIOP aerosol extinctions nearby the Pushan_NU site. Figure 10b shows the vertical distributions of the aerosol extinctions at 532 nm over one CALIOP orbit path near Pushan_NU site around 18:00 UTC on 3 November, and Figs. 10d, f, h, j, and l show the corresponding simulated ones of the five experiments. It is apparent that the FR experiment tends to

overestimate the aerosol extinctions with the centers located at altitudes of 1-2 km from 33°N and 124°E to 38°N and 126°E, while the experiments with CALIOP assimilation especially 4D-LETKF-24H-CALIPSO experiment rather than the 4D-LETKF-24H-NRL experiment correctly eliminates the unrealistic high extinction coefficients, making both the aerosol extinctions and AOTs more in

accordance with the CALIOP and AERONET observations. This indicates that CALIOP assimilation is superior to the MODIS assimilation on the optimization of the aerosol vertical distribution, although the MODIS assimilation can better improve the aerosol total column properties.

**4.3 Independent Validation of the Analysis With Independent MODIS Aqua AOTs**

In order to perform an independent validation with the MODIS Aqua AOT products, we screen out the
portions of the MODIS Aqua AOT products which are same as the assimilated NRL MODIS dataset. As shown in Figs. 11a-j, we give the spatial distributions of the biases and RMSEs between the simulated and the independent MODIS Aqua observed AOTs at 550 nm for the five experiments. The AOTs over the ocean areas are generally underestimated in the FR experiment, and all the four assimilation experiments correctly increase the sea salt aerosols, leading to the lower biases and
RMSEs over the ocean. It is apparent that the four assimilation experiments also improve the model performances over the tropic Atlantic located at the downwind of North Africa. This is mainly caused by the reductions of the transported dust and OC aerosols (shown in Fig. S2), indicating both the CALIOP and MODIS assimilation improve the simulation of the characteristics of aerosol transport from Africa to Atlantic. For East China and Australia, the RMSEs in 4D-LETKF-24H-CALIPSO
experiment are obviously lower than the ones in LETKF-CALIPSO experiment.
Interestingly, it is found that the AOTs in the experiments with CALIOP assimilation have the larger biases and RMSEs with the MODIS Aqua AOTs over the western part of the Saharan desert compared to the FR experiment, whereas such discrepancy is not found between the FR and 4D-LETKF-24H-NRL experiments. The CALIOP (MODIS) assimilation reduces (increases) the dust AOTs over the
western part of the Saharan desert. To investigate the possible reason, Figure 12 gives the spatial distributions of the multi-annual mean AOTs at 550 nm for MODIS Aqua, day-time CALIOP, and night-time CALIOP AOTs at 532 nm. The CALIOP AOTs are significantly lower than the MODIS AOTs over the western Saharan desert region, indicating that the larger discrepancy of AOTs comparisons with the MODIS by the CALIOP assimilation is probably due to the differences between
the CALIOP and MODIS observations in this region. Ma et al. (2013) also mentioned that the CALIPSO AOT is significantly lower than the MODIS AOT over dust regions especially over the Saharan region. Schuster et al. (2012) found that the relative and absolute biases are probably due to the assumed Lidar ratio for the CALIPSO dust retrieval is too low. We also show the PDF plots of the AOT deviations between the simulated hourly AOTs and the independent MODIS Aqua observed ones
for the five experiments. Similar as the comparison with AERONET AOTs, the frequencies of AOT deviations within ±0.05 (±0.10) in 4D-LETKF-24H-CALIPSO is slightly higher than the ones in the LETKF-CALIPSO experiment. Moreover, the shape of the PDF plots in 4D-LETKF-24H-NRL and 4D-LETKF-24H-CALIPSO-NRL are similar except the peak of the experiment with multi-sensor assimilation is a bit lower than the other one.

## 4.4 Independent Validation of the Analysis With Independent CALIOP Aerosol Extinctions

To perform the independent validation of the aerosol vertical distribution, we conduct other three assimilation experiments same as the LETKF-CALIPSO, 4D-LETKF-24H-CALIPSO, and 4D-LETKF-24H-CALIPSO-NRL experiments except only assimilating the CALIOP observations in the night-time (hereafter called the LETKF-CALIPSO-night, 4D-LETKF-24H-CALIPSO-night, and 4D-LETKF-24H-CALIPSO-NRL-night), and then we use the remaining CALIOP observations in the day-time for independent validation. Since the CALIOP observations over the land in the day-time is very limited (shown in Fig. S3), we only show the results over the global ocean. Figures 13a-e show the scatter plots of the day-time CALIOP extinctions versus the simulated ones in the five experiments. It is found that all the assimilation experiments improve the model performance of the simulated aerosol vertical distribution. With respect to the CALIOP assimilation, the 4D-LETKT method is obviously superior to the LETKF. This is because the analyses are the same as the forecast results during the time slots when there are no CALIOP extinction observations to be assimilated in the LETKF experiment, since the LETKF only optimizes the aerosol fields when CALIOP extinction observations are available and provide the initial conditions for the next forward simulation. However, the 4D-LETKF can optimize the aerosol fields over all the time slots in the assimilation window by assimilating the asynchronous observations. The 4D-LETKF CALIOP assimilation is better than the that of MODIS assimilation, indicating the optimized aerosol vertical distributions are more benefited from the CALIOP vertical observations than the MODIS column intergraded observations. Based on the statistical metrics, the 4D-LETKF-24H-CALIPSO-NRL-night experiment has the best performance among the four assimilation experiments. This is probably due to the aerosol vertical distributions, which are unable to be optimized by assimilating the sparse CALIOP observations, are further optimized by the MODIS observations. The PDF plots in Figure 13f further prove the aerosol vertical observations are critical for constraining the aerosol vertical simulation and the simultaneous assimilation of CALIOP and MODIS observations has the best performance. It is obvious that the aerosol extinctions in the day-time is underestimated in the FR experiment over the global ocean, whereas the underestimation is improved by both MODIS and CALIOP assimilations and the peak is nearer to 0. Merely 11.31% (24.84%) of the extinction deviations in the FR experiment are within $\pm0.01(\pm0.02)$, whereas 35.01% (57.95%) of the extinction deviations in the 4D-LETKF-24H-CALIPSO-NRL-night experiment are within $\pm0.01(\pm0.02)$.

As shown in Fig.14, we also give the mean vertical profiles of the CALIOP observed and the model simulated aerosol extinction coefficients over the global ocean in the day-time. We ignore the levels above 2 km since there are limited CALIOP observations. It is apparent that there are significant discrepancies between the FR experiment and the CALIOP observation. Both the CALIPSO assimilation with the LETKF and the MODIS assimilation have limited improvements of the simulated aerosol extinction profiles. With respect to the 4D-LETKF-24H-CALIPSO-night and the 4D-LETKF-24H-CALIPSO-NRL-night experiments, the aerosol profiles below 0.6 km are generally consistent with the independent CALIOP observations, and the aerosol profiles from 0.6 km to 1.2 km are more comparable to the observed one than those of the other experiments. The discrepancies of the aerosol extinction profiles between the 4D-LETKF-24H-CALIPSO-night and 4D-LETKF-24H-CALIPSO-

NRL-night experiments are obvious above 1 km. This is because there are less CALIOP observations over 1 km, and this induces the NRL MODIS AOTs play a more important role in modifying the profile.

From our results so far, both the CALIOP and MODIS assimilation can improve the magnitude of the simulated aerosol extinctions, however the CALIOP assimilation is superior to the MODIS assimilation in terms of modifying the incorrect aerosol vertical distributions and reproducing the real magnitudes and variations. The simultaneous assimilation of the CALIOP and MODIS observations is better than the separate assimilation in reproducing the aerosol vertical information.

## 5 Discussions

CALIPSO provides sparse observations due to the sensor-specific data gaps for a 16-day repeat cycle. To investigate the effects of the CALIPSO sensor gaps on the data assimilation, Fig. 15 shows the assimilation efficiencies (AE), referred to Yumimoto and Takemura (2011), with the MODIS Aqua observed AOTs as a function of the CALIOP overpass time. The hourly and daily mean AEs both show decreasing trends as the increasing distances of the assimilation time. The AEs are high close to the overpass time but deteriorate later on. Such a deterioration demonstrates that more intensive vertical observations can improve the aerosol vertical assimilation efficiency and hence advance the studies of the aerosol effects on the climate and environment.

To investigate the effects of the assimilation system parameters (i.e., horizontal localization length, vertical localization length, and uncertainty of dust emission) for the CALIOP and MODIS assimilation in this study, other five assimilation experiments named 4D-LETKF-24H-CALIPSO-HL500, 4D-LETKF-24H-CALIPSO-HL1000, 4D-LETKF-24H-CALIPSO-HL2000, 4D-LETKF-24H-CALIPSO-VL, and 4D-LETKF-24H-CALIPSO-Dust are conducted. The experiments are in the same model configuration as that of the 4D-LETKF-24H-CALIPSO experiment except the horizontal localization lengths of 500 km, 1000km, and 2000km used in the 4D-LETKF-24H-CALIPSO-HL500, 4D-LETKF-24H-CALIPSO-HL1000, 4D-LETKF-24H-CALIPSO-HL2000 experiments, four times vertical localization length used in the 4D-LETKF-24H-CALIPSO-VL experiment, and five times assumed dust uncertainty used in the 4D-LETKF-24H-CALIPSO-Dust experiment. Figure 16 shows the frequencies of the AOT deviations between the simulations of the six CALIOP assimilation experiments and the independent MODIS observations. In addition, the corresponding result of the FR experiment is also shown as a reference. Among the six assimilation experiments, the 4D-LETKF-24H-CALIPSO experiment shows the highest percentages of deviations between ±0.05 and ±0.10. By the increments of the horizontal localization length, the percentages of deviations between ±0.05 and ±0.10 are decreasing and the peaks of the PDF plots tend to far away from 0. The significant differences between 4D-LETKF-24H-CALIPSO and 4D-LETKF-24H-CALIPSO-VL experiments indicate that one model layer is a reasonable vertical localization length and a larger one will reduce the assimilation ability. Therefore, we use the 200 km horizontal localization length (i.e., observations located within 730 km are assimilated), one model layer vertical localization length, and 100% dust perturbed assuming uncertainties for the aerosol data assimilation in this study.

**6 Conclusions**

Assimilation of aerosol vertical observations can provide more accurate spatio-temporal distributions of aerosol characteristics especially the vertical information, which should advance the studies of the aerosol effects on the earth system. In the present study, we develop the 4D-LETKF assimilation system for the CALIOP vertical extinction observations and successfully present an one-month long hourly aerosol analyses during November 2016. The hourly analyses are compared with the assimilated observations as the internal checks and validated by the independent AERONET retrieved AOTs, MODIS Aqua AOTs, and the day-time CALIOP extinctions. The effects of assimilating the observations including the vertical information or not and the impacts of the multi-sensor assimilation on the model performances are also investigated.

Compared with the assimilated CALIOP observations as the internal checks, the two CALIOP assimilation experiments are evidently better than the FR experiment especially over the ocean regions. Compared with the assimilated NRL MODIS AOTs, the 4D-LETKF-24H-NRL experiment is superior to the FR experiment as indicated by the reduced biases. These results indicate the assimilation system is operated successfully. The performances of the 4D-LETKF-24H-CALIPSO experiment are generally comparable to those of the LETKF-CALIPSO experiment. This indicates the weights determined at the end of the 24 hours assimilation window are valid to optimize the ensemble trajectories and the temporally remote asynchronous observations within 24 hours have limited influence on the analysis. Moreover, the elapsed time for the one-month assimilation over November 2016 with 4D-LETKF-24H-CALIPSO is much shorter than that of the LETKF-CALIPSO experiment. This is due to the 4D-LETKF can avoid frequent switching between the assimilation and model ensemble forecasts.

Compared with the independent MODIS and AEROENT retrieved AOTs, CALIOP and MODIS assimilation both can achieve improved model simulated AOTs over most of the land and ocean regions. However, the experiments with MODIS assimilation can reproduce better agreements with the independent AERONET AOTs than the experiments only assimilating CALIOP observations. This is probably due to the CALIOP (with a 16-day repeat cycle) has much sparser coverages than the MODIS and the assimilation efficiencies of CALIOP decreases with the increasing distances of the assimilation time. Compared with the individual MODIS assimilation, the inclusion of the CALIOP observations has an insignificant impact on the AOT analysis. It is because the MODIS observations include more aerosol column information than the CALIOP observations.

With assimilating the night-time CALIOP observations and using the remaining CALIOP observations in the day-time for independent validation, it is found that both the CALIOP and MODIS assimilation can improve the magnitude of the simulated aerosol extinctions, however the CALIOP assimilation is superior to the MODIS assimilation in terms of modifying the incorrect aerosol vertical distributions and reproducing the real magnitudes and variations. Compared with the independent CALIOP extinction observations in the day-time over the ocean, the 4D-LETKF CALIOP assimilation is better than that of the MODIS assimilation. It indicates the optimized aerosol vertical distributions are more benefited from the CALIOP vertical observations than the MODIS column intergraded observations. The simultaneous CALIOP and MODIS assimilation experiment has the best performance. This is

probably due to the aerosol vertical distributions, which are unable to be optimized by assimilating the sparse CALIOP observations, are further optimized by the MODIS observations.

The assimilation of the CALIOP extinction observations deteriorates the model performance over the western part of the Sahara desert compared to the MODIS observations. This is due to the inconsistences between the CALIOP and MODIS observations, indicating the assumed lidar ratio which is important for aerosol extinction retrieval still has the uncertainties especially in the dust regions and the data quality needed to be improved for advancing model skill over these regions.

The assimilation efficiencies are high close to the overpass time but deteriorate later on. This indicates that more aerosol vertical observation platforms are required to fill the sensor-specific observation gaps for better aerosol vertical data assimilation. In the near future, joint assimilation of the aerosol profile observations from the Earth Cloud Aerosol and Radiation Explorer (EarthCARE) and CALIPSO may further advance our understanding of the atmospheric aerosol vertical characteristics.

**Code/Data availability.** The data and data analysis method are available upon request. The NICAM source code is available with the terms and conditions in http://nicam.jp/hiki/?Research+Collaborations.

**Author contributions.** Tie Dai designed the experiments. Yueming Cheng carried out the experiments and conducted the data analysis with contributions from all co-authors. Yueming Cheng prepared the manuscript with help from Tie Dai, Daisuke Goto, Nick A.J. Schutgens, Guangyu Shi, and Teruyuki Nakajima.

**Competing interests.** The authors declare that they have no conflict of interest.

**Acknowledgments.** This study is financially supported by the Strategic Priority Research Program of the Chinese Academy of Sciences (XDA2006010302), the National Key R&D Program of China (2016YFC0202001, 2017YFC0209803), the National Natural Science Funds of China (41571130024, 41605083, 41590875, and 41475031). Some of the authors are supported by the Global Environment Research and Technology Development Fund S-12 of MOEJ, the Japan Aerospace Exploration Agency (JAXA)/Earth Observation Priority Research, and the Grant-in-Aid for Young Scientist A (grant 17H04711) in Japan. The model simulations were performed using the supercomputer resources NIES/NEC SX-ACE and JAXA/JSS2. We are grateful to the relevant researchers who provided the observation data of AERONET (http://www.eorc.jaxa.jp/ptree/index.html) and MODIS (https://modis-atmos.gsfc.nasa.gov/products/aerosol). CALIPSO data were obtained from the NASA Langley Research Center Atmospheric Sciences Data Center (ASDC). We also thank two anonymous reviewers for their insightful comments and suggestions.

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

# Figures

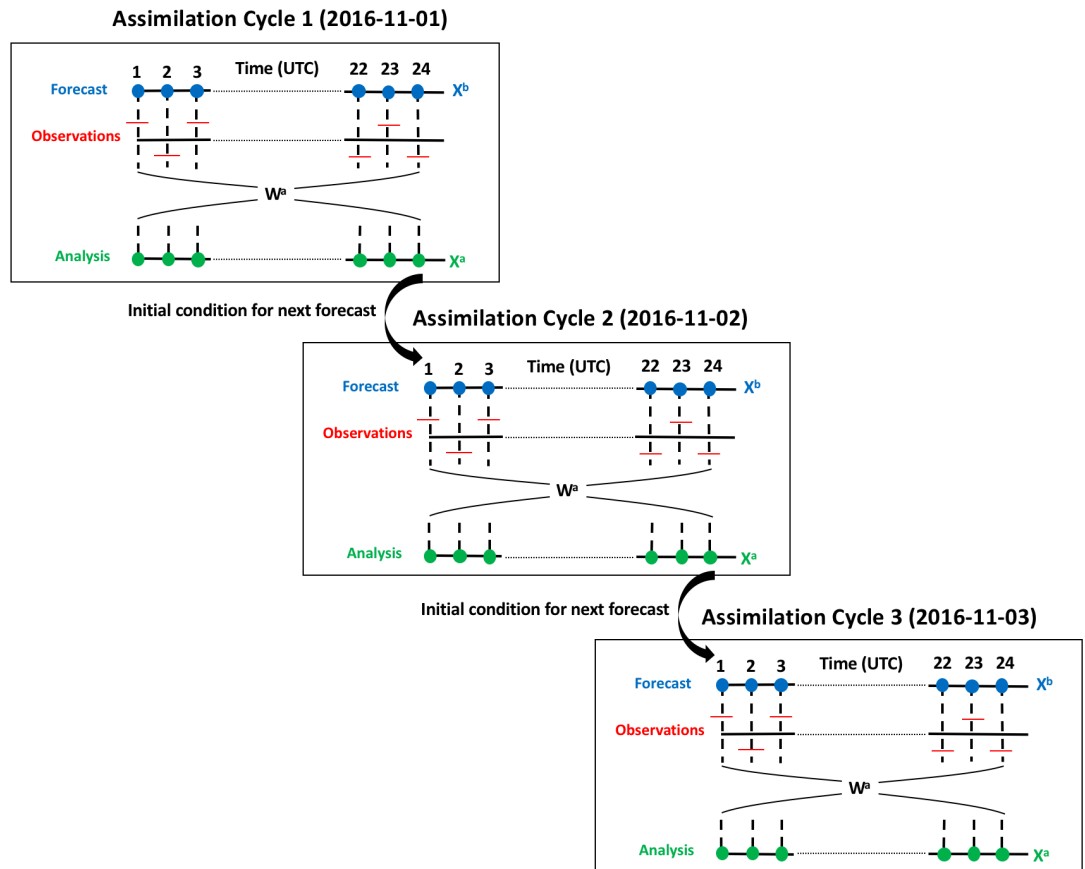

**Figure 1.** Schematic illustrating the three assimilation cycles of 4D-LETKF. Hourly observations are shown as red line segments for the each 24 hr assimilation cycle. The blue dots represent the 24 hr forecast that are hourly output. The green dots represent the hourly analysis within the 24 hr assimilation window.

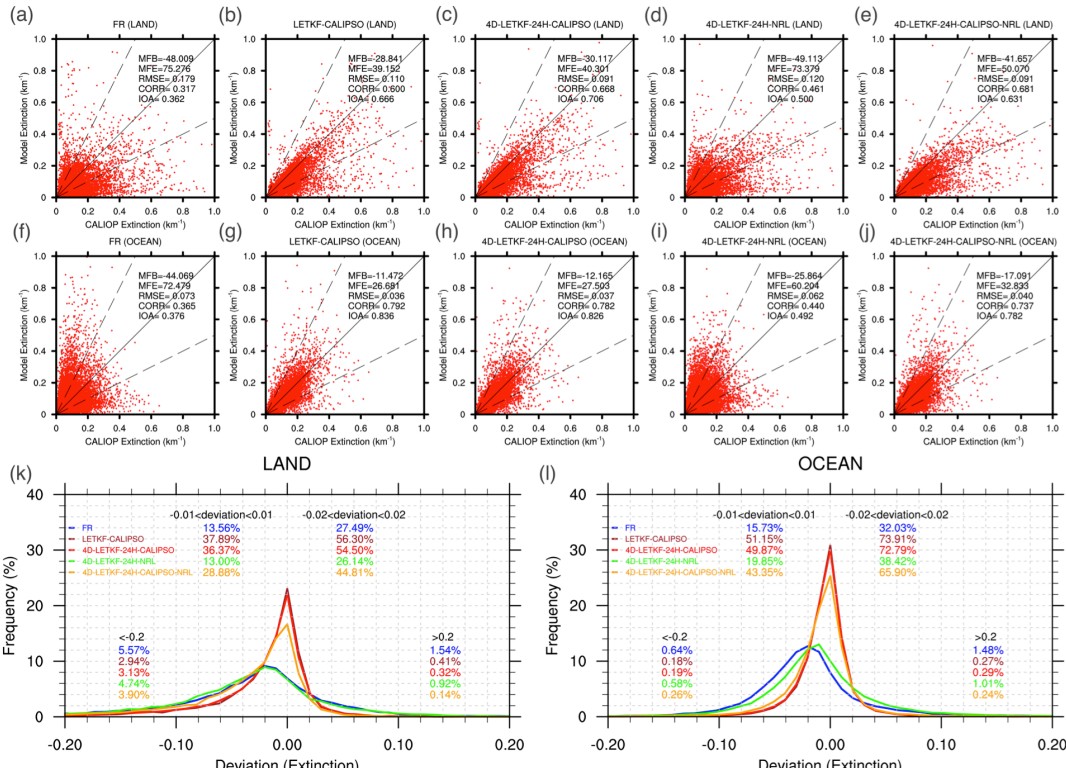

**Figure 2.** Scatter plots of the CALIOP (Cloud-Aerosol Lidar with Orthogonal Polarization) hourly aerosol extinction coefficients at 532 nm [km⁻¹] versus the simulated ones at 550 nm over the global land (a, b, c, d, e) and ocean (f, g, h, i, j) during November 2016 for the FR, LETKF-CALIPSO, 4D-LETKF-24H-CALIPSO, 4D-LETKF-24H-NRL, and 4D-LETKF-24H-CALIPSO-NRL experiments. The solid black line is the 1:1 line and the dashed black lines correspond to the 1:2 and 2:1 lines. MFB, MFE, RMSE, CORR, and IOA represent the mean fractional bias, the mean fractional error, the root mean square error, the correlation coefficient and the index of agreement, respectively. Frequency distributions of deviations (modeled extinction coefficients minus the CALIOP observed ones) over the global land (k) and ocean (l). The percentages of deviations between ±0.01, ±0.02, <-0.2 and >0.2 are also shown.

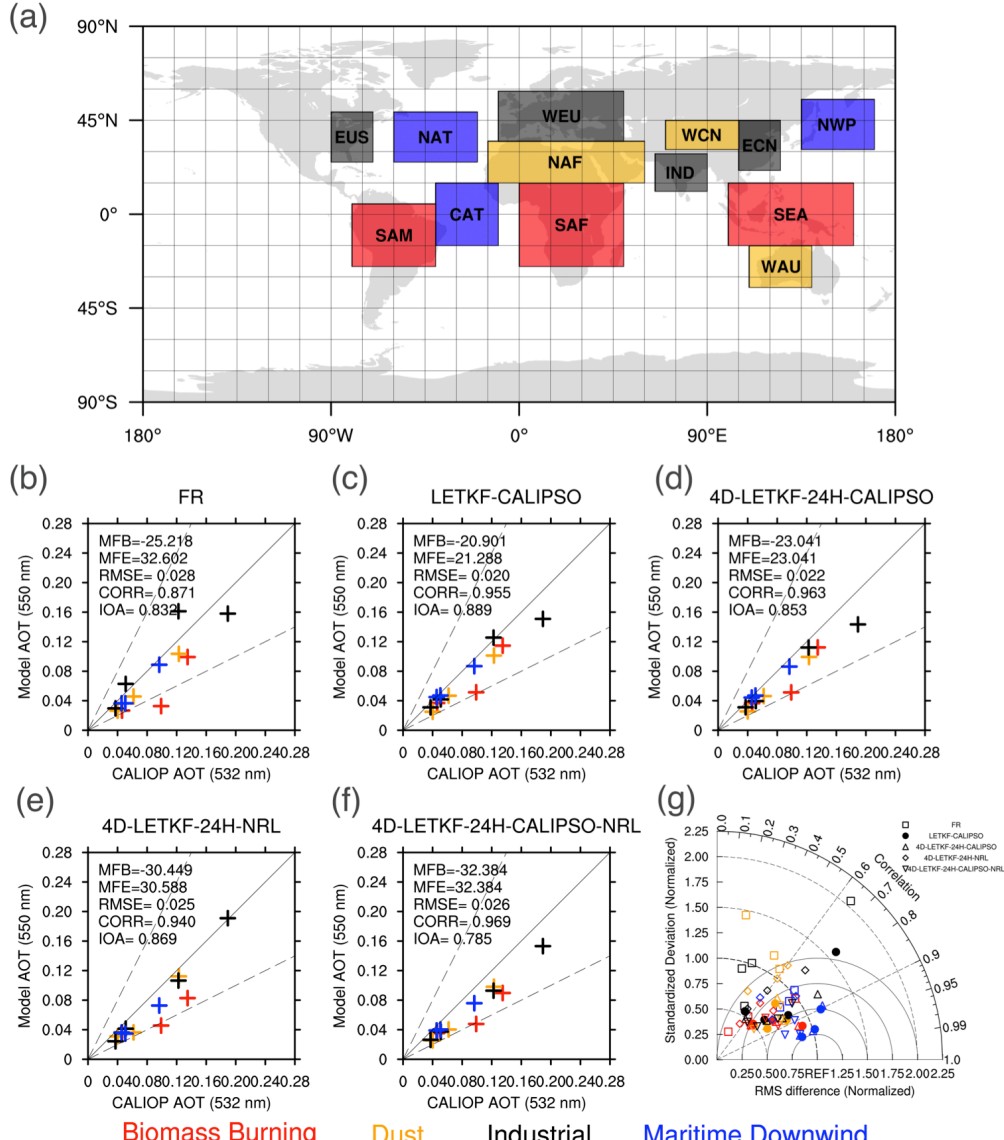

**Figure 3.** (a) Thirteen domains selected for regional analysis in this study. The red, orange, black and blue boxes indicate source regions of biomass burning smoke (SAM, SAF, and SEA), dust (NAF, WCN, and WAU), industrial pollution (IND, ECN, WEU, and EUS), and the outflow maritime regions downwind of major dust and industrial pollution sources (NWP, NAT, and CAT), respectively. Scatter plots of the simulated regionally averaged monthly mean column-integrated aerosol extinctions at 550 nm versus the collocated CALIOP ones at 532 nm over the thirteen selected regions during November 2016 for the FR (b), LETKF-CALIPSO (c), 4D-LETKF-CALIPSO (d), 4D-LETKF-24H-NRL (e), and 4D-LETKF-24H-CALIPSO-NRL (f) experiments. (g) The Taylor graph describing the column-integrated aerosol extinctions in the five experiments compared with the observed ones.

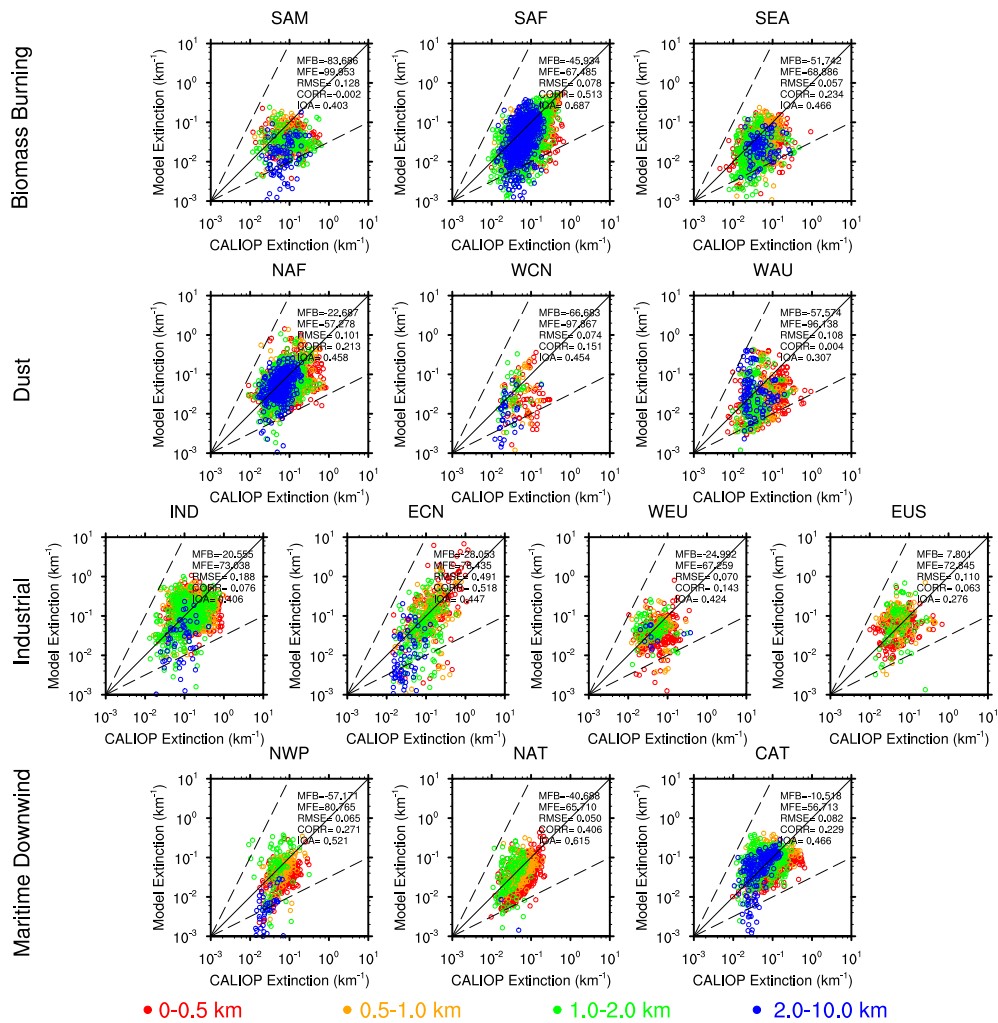

**Figure 4.** Scatter plots of the hourly modeled aerosol extinction coefficients at 550 nm [km⁻¹] versus the CALIOP observed ones at 532 nm over the thirteen selected regions during November 2016 for the FR experiment.

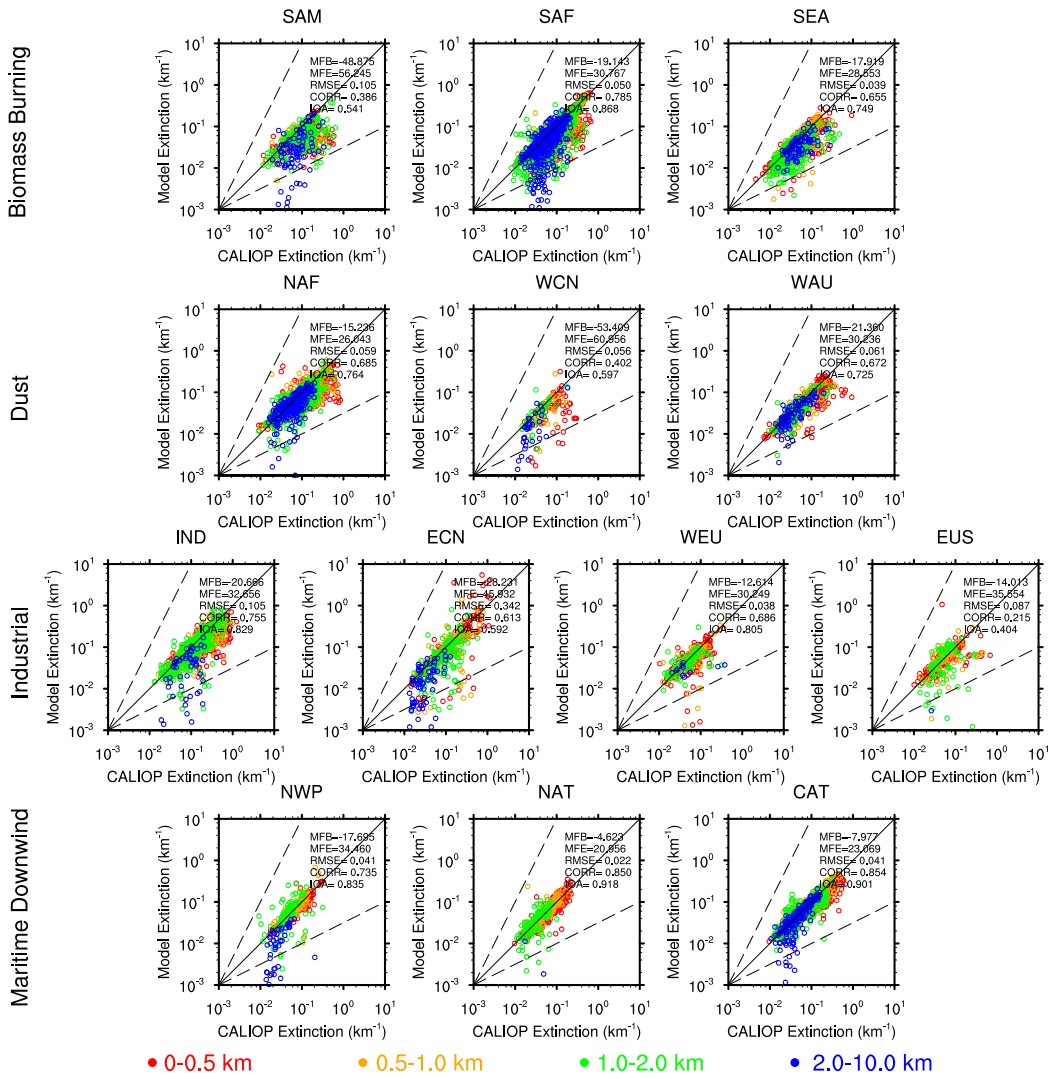

**Figure 5.** Same as Figure 4 but for the LETKF-CALIPSO experiment.

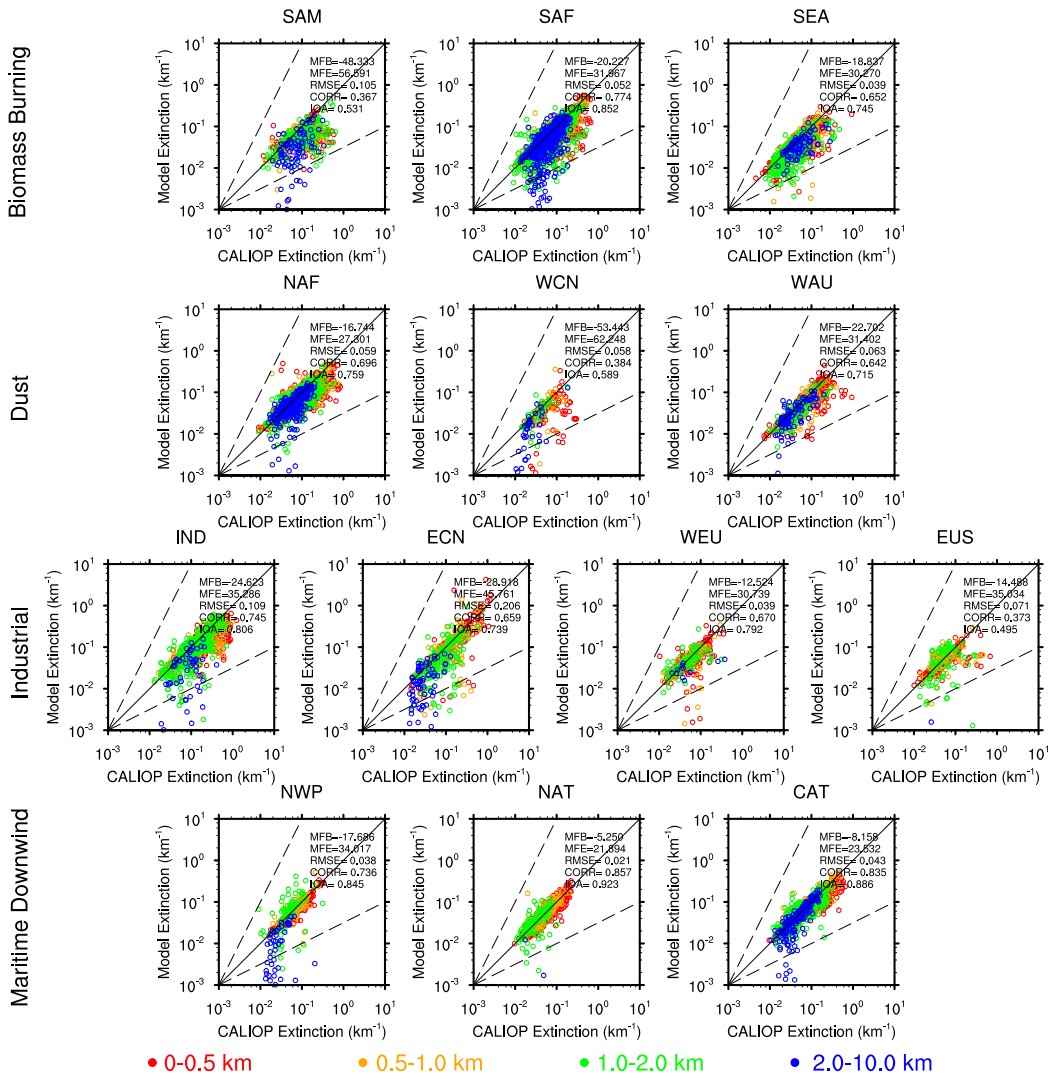

**Figure 6.** Same as Figure 4 but for the 4D-LETKF-24H-CALIPSO experiment.

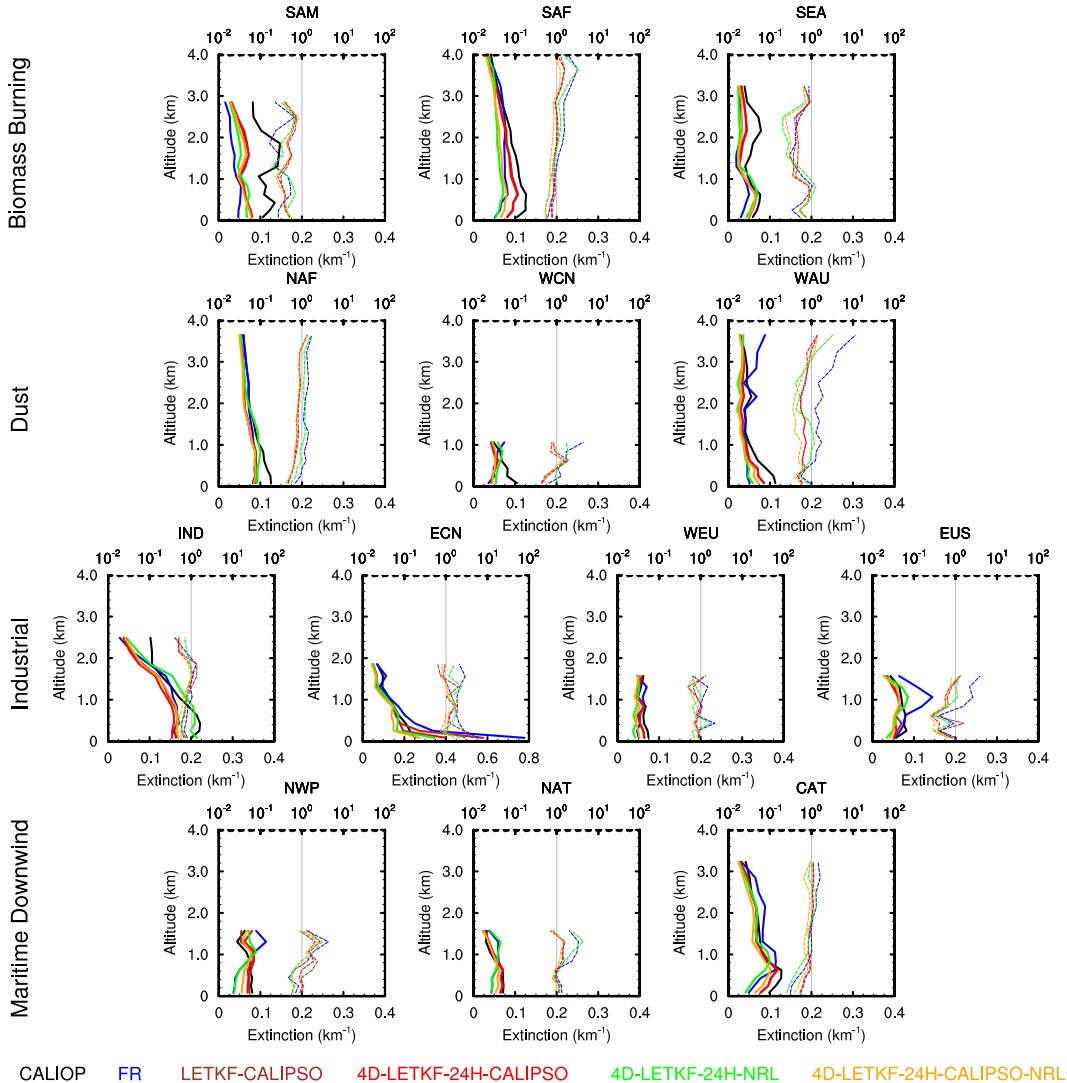

**Figure 7.** Regionally averaged monthly mean vertical profiles of the five experiments simulated aerosol extinction coefficients at 550 nm [km⁻¹] and the CALIOP observed ones at 532 nm (solid lines) and the ratios between the simulated and observed standard deviations (dashed line) over the thirteen selected regions during November 2016.

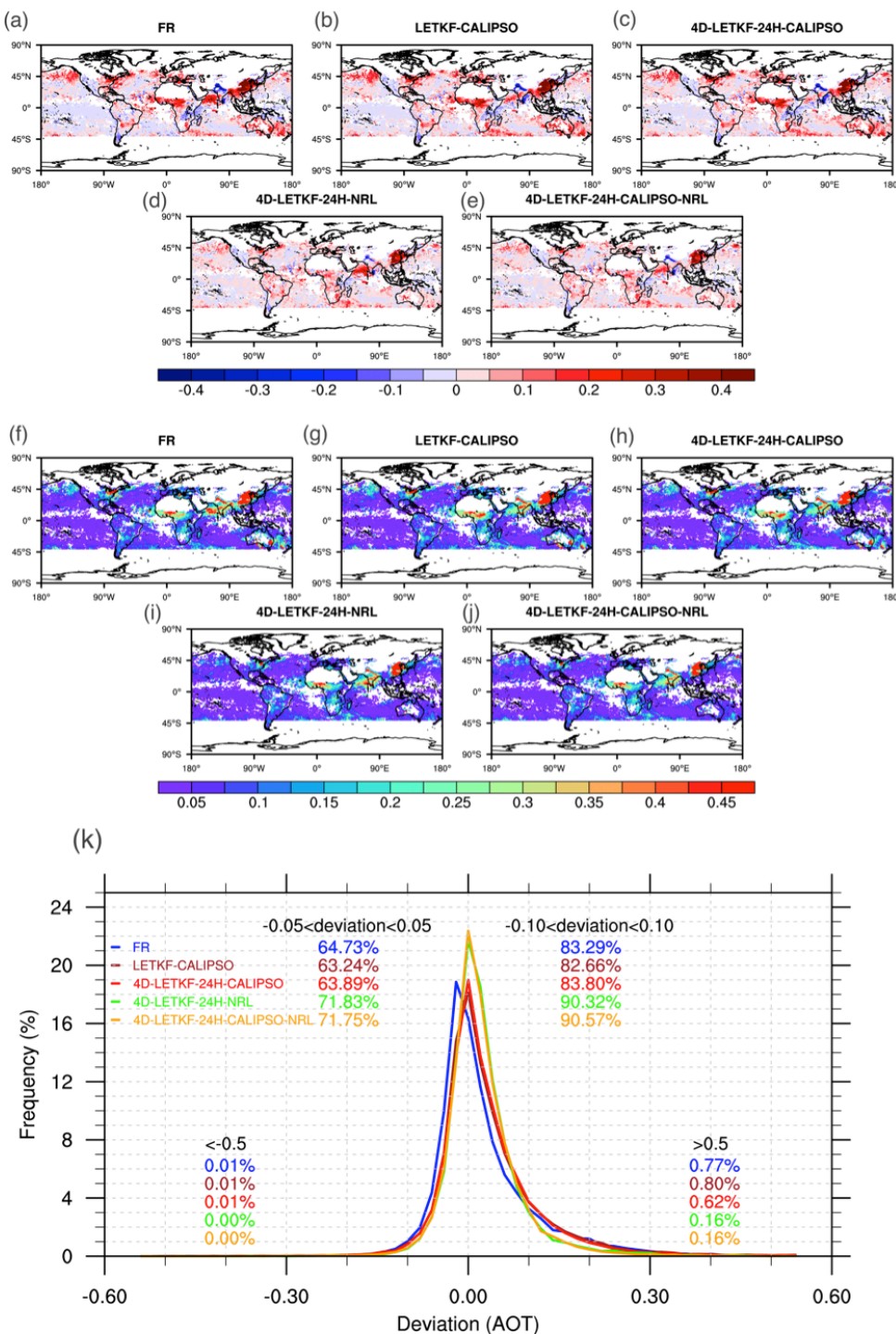

**Figure 8.** Spatial distributions of the biases (the simulated AOTs minus the observed ones) (a, b, c, d, e) and root-mean-square errors (f, g, h, i, j) between the simulated and the Naval Research Laboratory (NRL) Moderate Resolution Imaging Spectroradiometer (MODIS) AOTs at 550 nm during November 2016 for the FR, LETKF-CALIPSO, 4D-LETKF-24H-CALIPSO, 4D-LETKF-24H-NRL, and 4D-LETKF-24H-CALIPSO-NRL experiments. (k) Frequency distributions of

deviations (the simulated AOTs minus the observed ones) from the NRL MODIS observations. The percentages of deviations between ±0.05, ±0.10, <-0.5 and >0.5 are also shown.

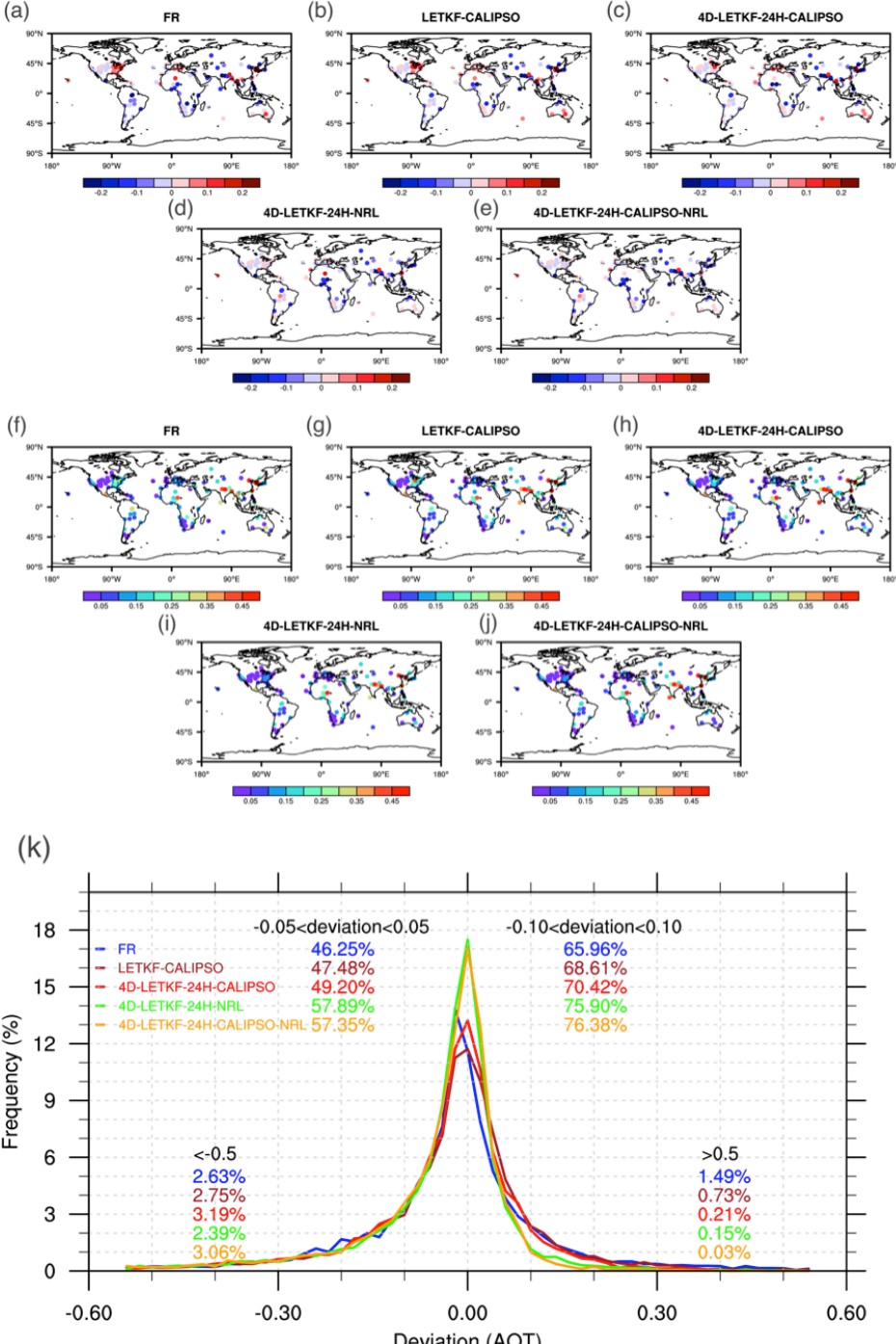

 **Figure 9.** Same as Figure 8 but for the simulated AOTs at 440 nm against the AErosol RObotic NETwork (AERONET)-retrieved ones.

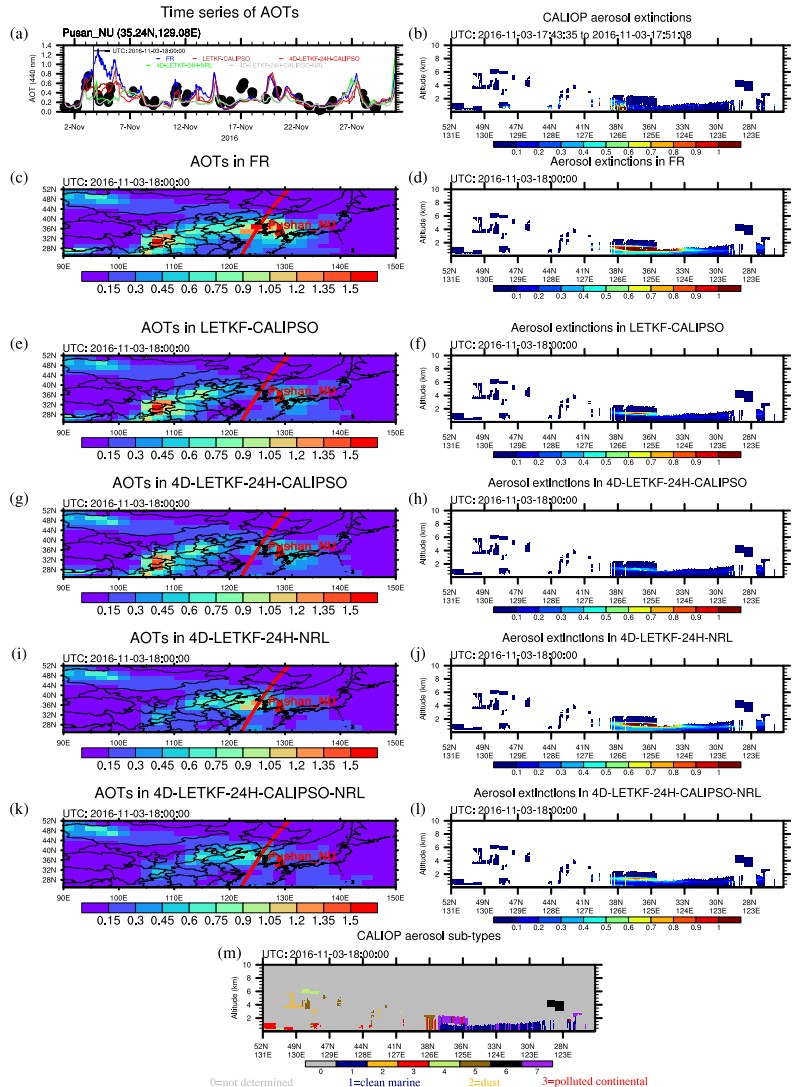

**Figure 10.** (a) Hourly time series of the AOTs at 440 nm for the FR, LETKF-CALIPSO, 4D-LETKF-24H-CALIPSO, 4D-LETKF-24H-NRL, and 4D-LETKF-24H-CALIPSO-NRL experiments and the observed AOTs from AErosol RObotic NETwork (AERONET) over
5   Pushan_NU site during November 2016. The root-mean-square error (RMSE) and correlation coefficient (CORR) between the simulated and the observed AOTs are also shown. The spatial distributions of the simulated AOTs at 440 nm at 18:00:00 (UTC) 3 November 2016 in the FR (c), LETKF-CALIPSO (e), 4D-LETKF-24H-CALIPSO (g), 4D-LETKF-24H-NRL (i), and 4D-LETKF-24H-CALIPSO-NRL (k) experiments. The red triangle indicates the location of
10   Pushan_NU site. The red curve indicates one CALIPSO orbit path near the Pushan_NU site on 3 November 2016. The CALIOP observed aerosol extinction coefficients at 532 nm [km$^{-1}$] (b) and the simulated ones at 550 nm in the FR (d), LETKF-CALIPSO (f), 4D-LETKF-24H-CALIPSO (h), 4D-LETKF-24H-NRL (j), and 4D-LETKF-24H-CALIPSO-NRL (l) experiments over that CALIPSO orbit path. (m) CALIPSO derived vertical aerosol sub-types.

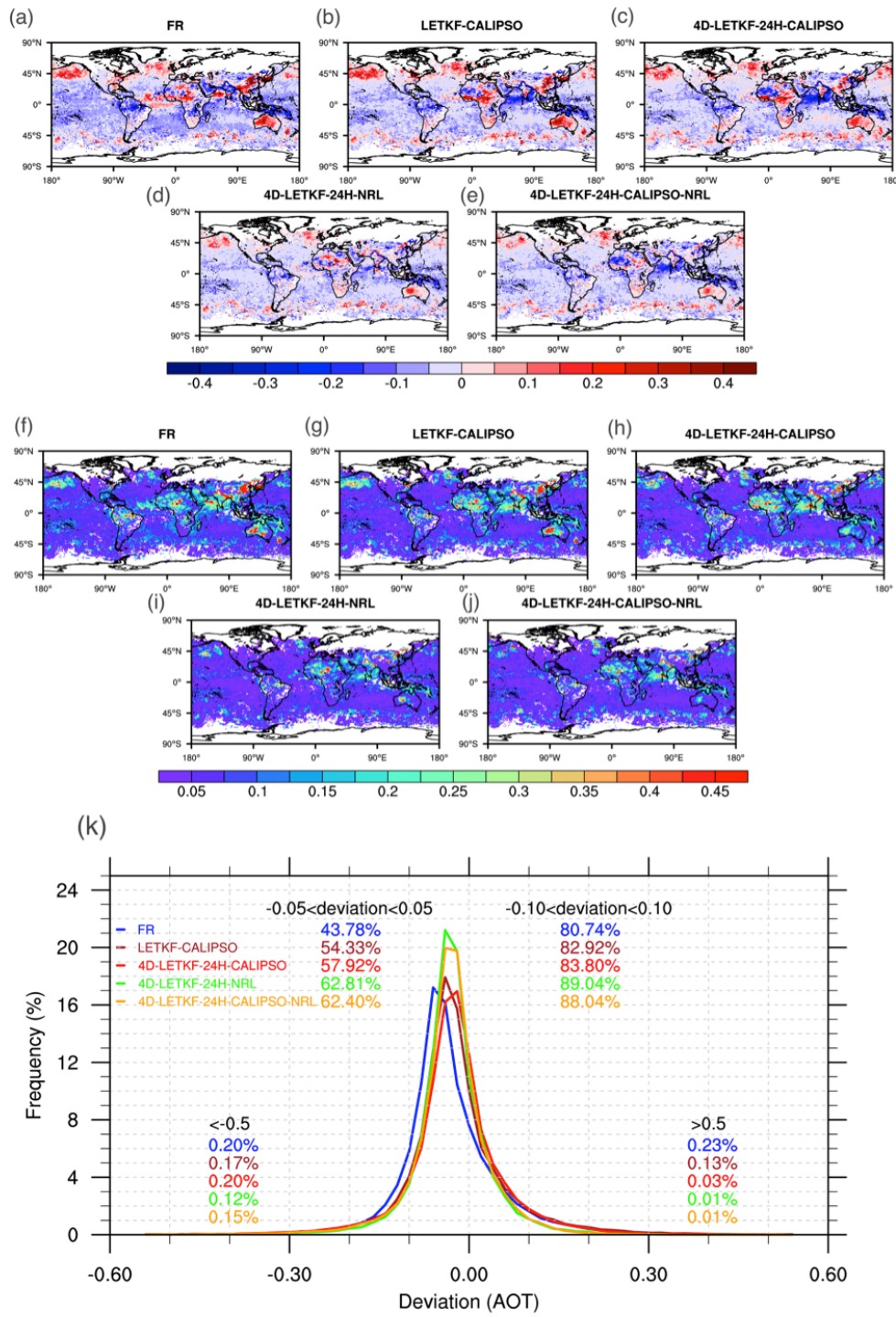

**Figure 11.** Same as Figure 8 but for the simulated AOTs at 550 nm against the independent MODIS Aqua ones.

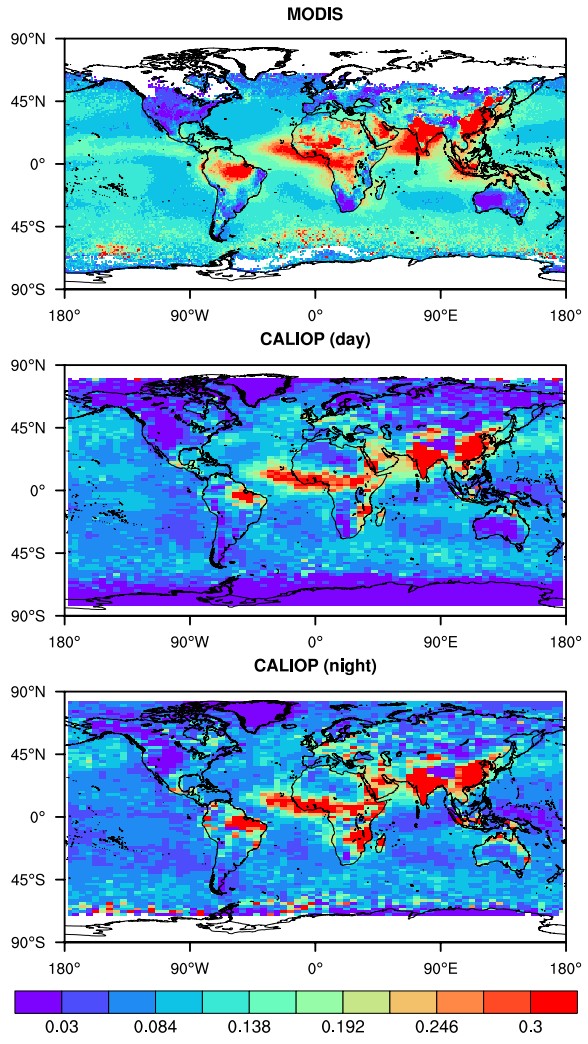

**Figure 12.** Spatial distributions of the monthly mean MODIS Aqua AOTs at 550 nm, day-time CALIOP and night-time CALIOP AOTs at 532 nm in November from 2006 to 2016.

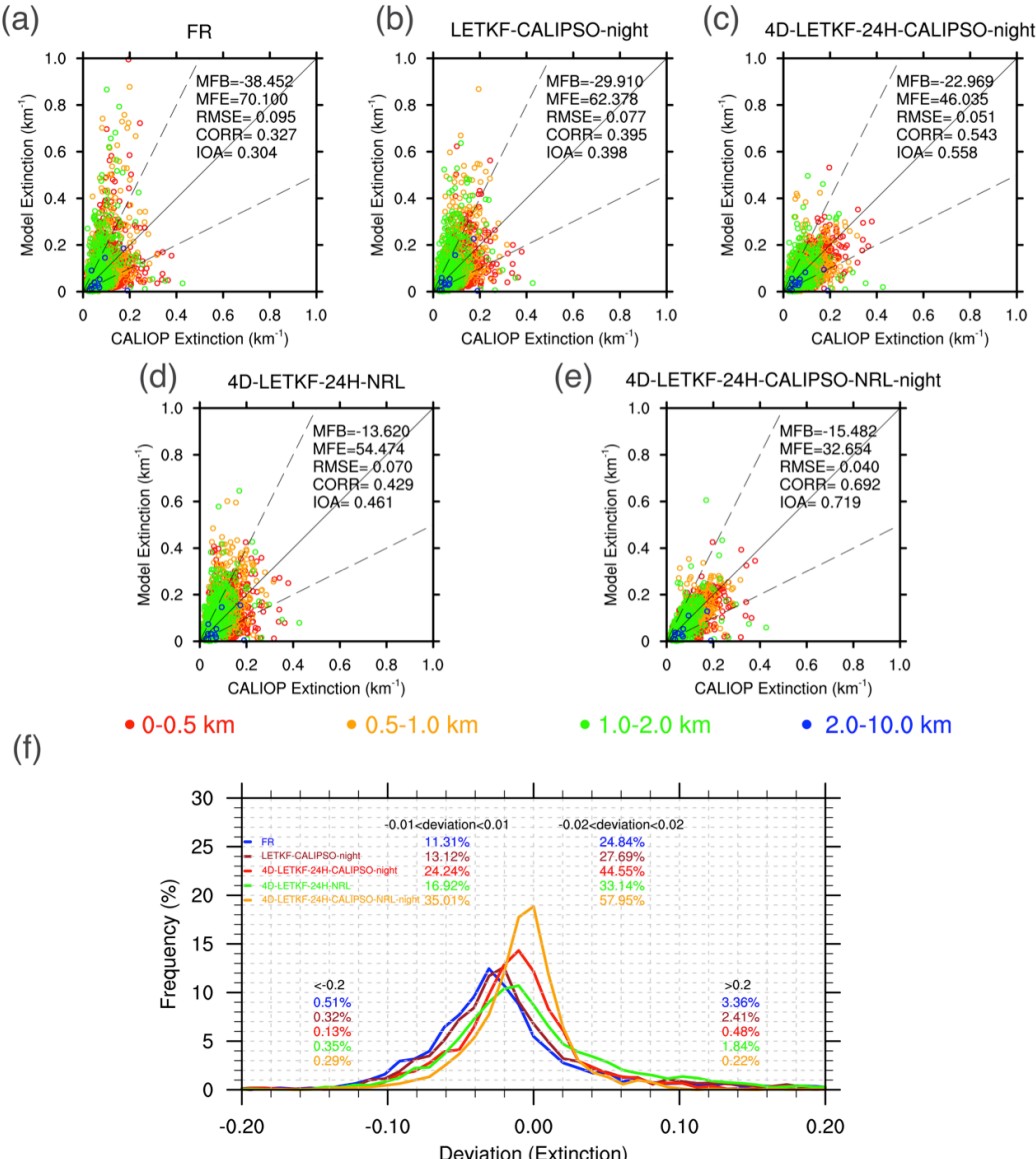

**Figure 13.** Scatter plots of the simulated hourly aerosol extinction coefficients in the day-time at 550 nm [km⁻¹] for the FR (a), LETKF-CALIPSO-night (b), 4D-LETKF-24H-CALIPSO-night (c), 4D-LETKF-24H-NRL (d), and 4D-LETKF-24H-CALIPSO-NRL-night (e) experiments versus the CALIOP observed ones in the day-time at 532 nm. (f) Frequency distributions of deviations (modeled extinction coefficients minus the CALIOP observed ones in the day-time). The percentages of deviations between ±0.01, ±0.02, <-0.2 and >0.2 are also shown.

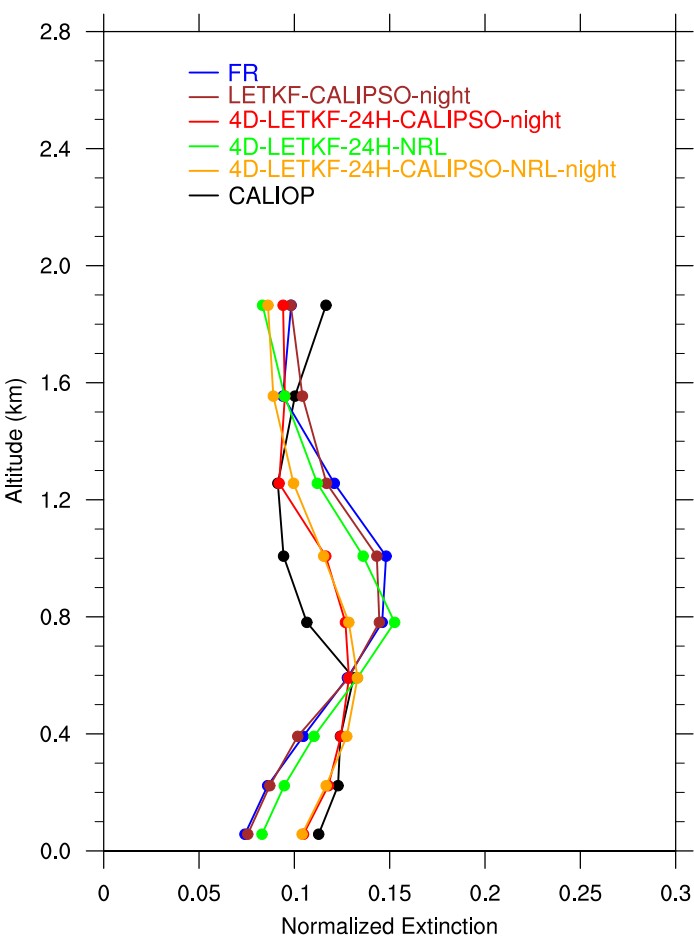

**Figure 14.** Averaged monthly mean vertical profiles of the simulated normalized aerosol extinction coefficients in the day-time at 550 nm [km$^{-1}$] in the five experiments and the CALIOP observed ones in the day-time at 532 nm.

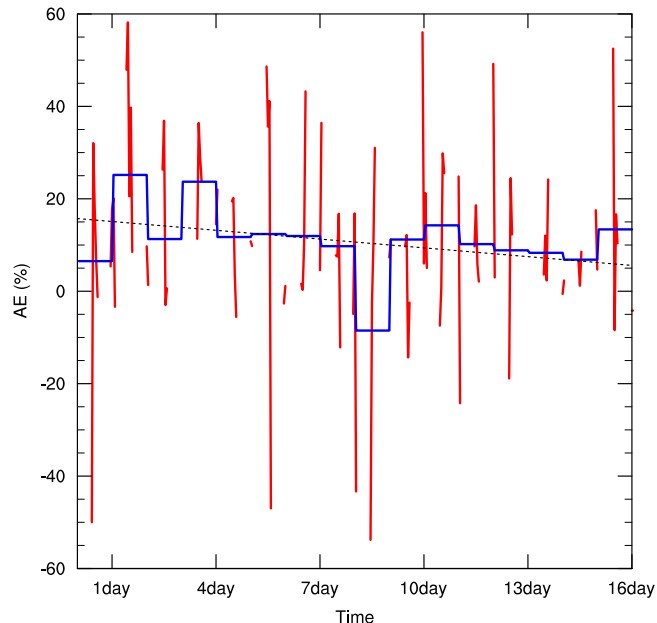

**Figure 15.** Assimilation efficiencies (AE) calculated against the Moderate Resolution Imaging Spectroradiometer (MODIS) Aqua observed AOTs at 550 nm for the 4D-LETKF-24H-CALIPSO experiment as the function of the distance of the assimilation time. The daily mean time series and the linear regression line are shown as the blue solid and the black dashed lines, respectively.

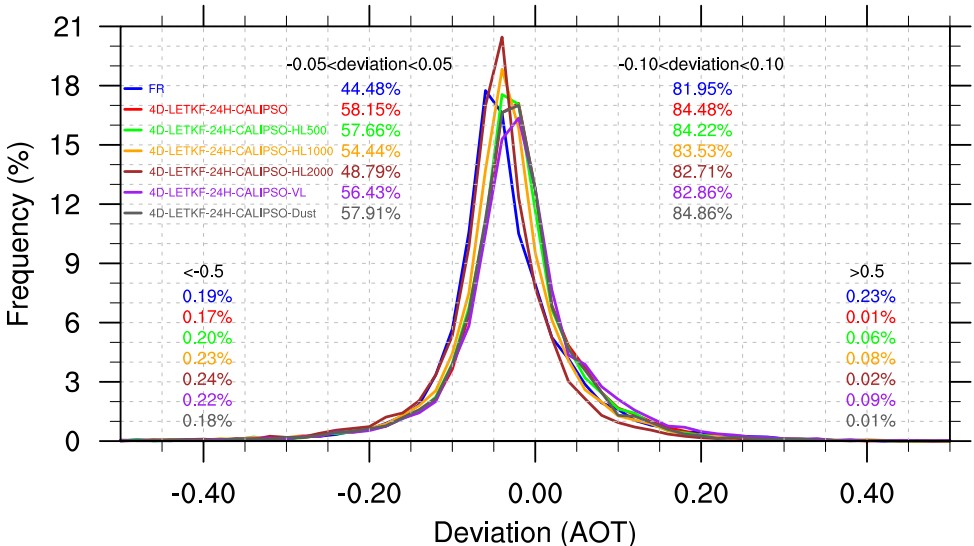

**Figure 16.** Frequency distributions of deviations (the simulated AOTs minus the observed AOTs) from the Moderate Resolution Imaging Spectroradiometer (MODIS) Aqua observed AOTs at 550 nm for the FR, 4D-LETKF-24H-CALIPSO, 4D-LETKF-24H-CALIPSO-L500, 4D-LETKF-24H-CALIPSO-L1000, 4D-LETKF-24H-CALIPSO-L2000, 4D-LETKF-24H-CALIPSO-VL, and 4D-

10 LETKF-24H-CALIPSO-Dust experiments. The percentages of deviations between ±0.05, ±0.10, <-0.5 and >0.5 are also shown.

**Tables**

**Table 1.** Experimental design for the sensitivity tests in this study

| Sensitivity experiments |
| --- |
| FR: single deterministic forecast without aerosol data assimilation |
| LETKF-CALIPSO: assimilates the CALIPSO vertical extinction coefficients and the cycle frequency and forecast length are set to 1 hr |
| 4D-LETKF-24H-CALIPSO: same as LETKF-CALIPSO except the cycle frequency and forecast length are set to 24 hr with hourly model output and analysis |
| 4D-LETKF-24H-NRL: same as 4D-LETKF-24H-CALIPSO except assimilates the NRL MODIS level-3 gridded AOTs |
| 4D-LETKF-24H-CALIPSO-NRL: same as 4D-LETKF-24H-CALIPSO except assimilates both the CALIPSO vertical extinction coefficients and NRL MODIS level 3 gridded AOTs |

