# Peer review of "Investigating the assimilation of CALIPSO global aerosol vertical observations using Four-Dimensional Ensemble Kalman Filter"

_Atmospheric Chemistry and Physics, 2019_

## Referee Comment (RC1) · Anonymous Referee #1 · 30 Jun 2019

General comments: In this manuscript, the authors applied the 4D-LETKF and an aerosol model SPRINTARS online coupled with NICAM to generate hourly aerosol horizontal and vertical analyses for one-month using the CALIOP aerosol extinctions. The results are validated using both the CALIOP extinctions and the MODIS and AERONET AOTs observations, and the assimilation experiments are also evaluated in a statistical sense. Some interesting results have been found. It is probably the first study to conduct the hourly aerosol vertical extinctions assimilation using four-dimensional ensemble Kalman method for one month. The manuscript is well written and logically organized in its structure. I recommend to publish it after minor revision.

[Figure]

Specific comments: 1. In Section 2, all the data used in this paper should be described. For example, the CALIPSO level 2 VFM product in the 'quality control procedures', the CALIPSO level 3 monthly mean gridded aerosol profile products in Figure 9.

2. In the DA-CALIPSO experiment, the CALIPSO vertical extinction coefficients are assimilated. Also, the CALIOP extinction coefficients serve as the reference standard in evaluating the CALIOP assimilation result (Fig. 1a-f, Figs. 3-5). The question is, how to avoid the uncertainties of CALIPSO product itself in the conclusion 'The CALIOP assimilation is superior to the MODIS assimilation in modifying the incorrect aerosol vertical distributions and reproducing the real magnitudes and variations'? So does the MODIS observation (Figs. 7-8).

3. Lines 349-357, Figure 9, The analysis should be repeated using CALIPSO Version 4 data, released last year, as the dust detection in Version 3 algorithms has been improved in Version 4. 1) In Version 3, any layer detected on single shots is classified as cloud. In Version 4, CAD is now applied to layers detected on single-shots. 2) The Version 3 CAD algorithm tends to classify elevated dust as cloud. This tendency is reduced in Version 4. The results 'the CALIOP AOTs are significantly lower over the western Saharan desert region' may be improved.

Technical corrections: 1. Lines 14-15, Add 'the' before 'Cloud-Aerosol Lidar with Orthogonal Polarization' and 'Cloud–Aerosol Lidar and Infrared Pathfinder Satellite Observation'.

2. Line 43, Please give the full spelling when the abbreviation first appears. POLDER.

3. Line 44, Please give the full spelling when the abbreviation first appears, MODIS.

4. Line 55, Please give the full spelling when the abbreviation first appears, NIES.

5. Line 91, 'Moderate Resolution Imaging Spectroradiometer (MODIS)' should be changed to 'MODIS'.

6. Lines 108-109, 'Takemura et al., 2000; Takemura et al., 2003; Takemura et al., 2009'

-> 'Takemura et al., 2000, 2003, 2009' 'Satoh et al., 2005; Satoh et al., 2008; Satoh et al., 2014' -> 'Satoh et al., 2005, 2008, 2014'

7. Line 140, 'the horizontal and the vertical localization factors' should be changed to 'the horizontal and vertical localization factors'.

8. Line 147, 'over November 2016' should be changed to 'in November 2016'.

9. Line 337, the grammatical error exists in 'the both the two assimilations correctly...'

10. Line 365, 'the South and East Asia' should be changed to 'South and East Asia'

11. Line 368, 'both the CALIOP and the MODIS assimilations' should be changed to 'both the CALIOP and MODIS assimilations'

---

## Referee Comment (RC2) · Anonymous Referee #2 · 4 Jul 2019

General comments:

The authors present the result of an aerosol analysis comparison between CALIOP/CALIPSO data assimilation and NRL-MODIS data assimilation using a 4-dimensional ensemble Kalman filter with a global aerosol model to demonstrate the effects of the vertical profile information on the analysis performance. The manuscript looks well written and organized in its structure. Some interesting results are included for data assimilation researchers. However, it is impugnable that the "hourly" aerosol vertical analysis is really scientifically plausible, although the authors claim the hourly analysis as the novelty of this paper. In addition, some of the validation processes the

authors used were not suitable to validate data assimilation results. Therefore, I do not recommend to publish this manuscript as an article of EGU's high-IF journals.

Specific comments:

(1)

The authors insist that this is the first study to conduct the hourly aerosol analysis using a four-dimensional data assimilation method. That is because an hourly data assimilation with a 24-hour assimilation window is mathematically inadequate, and thus, nobody has conducted it.

If I am not mistaken, the authors assimilate one observation again and again (probably 25 times) because the data assimilation is performed every hour and each time window is set 24 hours. In contrast, if the authors define the 1, 2, … 23-hour forecasts as the analysis, which means the data assimilation is done once per day, the authors' method is completely the same as the previous studies.

Theoretically, observations should be used strictly only once in the data assimilation process. Of course, in reality, a multiple use of one observation sometimes yields a good analysis result. However, those practical methods are often controversial. At least, it is not scientifically ingenious. The authors did not discuss this controversial issue at all in the manuscript. For example, Eugenia Kalnay's "Running in Place (RIP)" scheme (https://doi.org/10.1002/qj.652), which is slightly similar to the authors' method, assimilates one observation repeatedly. Therefore, Kalnay et al. carefully discussed the RIP scheme and its application range in their papers. Despite that, the RIP scheme has been controversial.

If the authors like to generate hourly aerosol analyses with hourly observations, a three-dimensional ensemble Kalman filter should be used to avoid a multiple use of one observation, or very narrow time localization should be applied to a four-dimensional ensemble Kalman filter to reduce the influence of temporally remote observations. In

that case, a 24-hour assimilation window is too long.

(2) Sections 2.2 and 3.2

Although the MODIS-Aqua DT/DB dataset is described in Section 2.2, the NRL MODIS dataset is not. Although it is described in Section 3.2 how to make super observations of CALIOP/CALIPSO, those of NRL MODIS are not described. Please clarify them. The NRL MODIS dataset is probably 6-hourly. If so, how did you make the hourly analysis of DA-NRL?

(3) Lines 167-168

Because the 4D-LETKF is not a smoother, the analysis has only one timeslot during its time window, which can be arbitrarily specified. When is the analysis timeslot during the 24-hour window in this study?

The control variables are fine/coarse mass mixing ratios only? The authors did not separately control all the prognostic variables of NICAM-SPRINTARS? Please describe it in detail.

(4) Line 181

The horizontal and vertical localization scales are described here, but the temporal localization scale is not. If the temporal localization is not applied, please specify it.

(5) Lines 241-246

The authors say "... the DA-CALIPSO experiment achieves more higher correlations and lower RMS differences than those with the MODIS assimilation. This indicates that ..., while the CALIOP assimilation can also improve the temporal and spatial variations of the aerosol extinctions," but I do not agree. The authors validated the two experiments with the CALIOP data that were assimilated only for the DA-CALIPSO experiment. The reason of the high correlation and lower RMSD of the DA-CALIPSO is the self-verification between the DA-CALIPSO results and the CALIOP data, which

should be strongly refrained for data assimilation validation. What the authors did in Section 4.1 was sanity or internal checks, not validation.

(6) Line 252

The DA-CALIPSO is not "clearly" superior to the FR and DA-NRL because in this section the authors merely checked the sanity of the DA-CALIPSO experiment. If the authors like to indicate the superiority of the DA-CALIPSO or DA-NRL results, independent observations should be used. In other way, the forecasts from the data assimilation analyses can be validated by the assimilated observations.

(7) Lines 260-263

Does this mean that the vertical data assimilation breaks the aerosol profile balance? The analysis of the aerosol profile becomes not similar to either the prior or observation information? This is very interesting.

(8) Line 281

The word "improvement" is not appropriate. The authors were simply tested the similarity between the DA-CALIPSO results and the assimilated observations in this section.

(9) Line 296

As the authors say, the MODIS and CALIOP observations are generally inconsistent. Therefore, the authors have to compare the MODIS and CALIOP observations used in this study before the validation of data assimilation results.

(10) Lines 321-322

Nobody can conclude that the CALIOP assimilation is "superior" to the MODIS assimilation before the validation with independent observations. In this section, the authors checked only the similarity between the CALIOP assimilation results and the assimilated CALIOP observations.

[Figure]

(11) Section 4.1

Section 4.1 is too long in comparison to Sections 4.2 and 4.3. The most important result is Section 4.3. Although Section 4.1 is much longer than Sections 4.2 and 4.3, it does not contain the similarity check between the data assimilation results and the NRL-MODIS data.

(12) Line 325

The authors described a comparison with the MODIS Aqua C6.1 AOT products as an independent validation. Probably it is false. The NRL MODIS dataset highly probably contains information very close to the MODIS Aqua C6.1 AOT products. If the authors like to insist that it is an independent validation, the portion originated from MODIS Aqua should be screened out before the NRL MODIS data assimilation.

(13) Line 326-328

I do not understand the necessity to eliminate the potential effects of the sparse CALIPSO observations. I think the authors would rather evaluate the effect of the CALIOP sparseness here.

(14) Line 346 and Line 347

It is not unexpectable that DA-CALIPSO has larger biases and RMSEs compared to FR. Generally, different remote instruments have large biases. Therefore, "deterioration" is a hasty conclusion. A careful evaluation of the MODIS AOT and CALIOP AOT must be done with independent observations other than Figure 9 to say "deterioration".

(15) Line 362 "The AERONET sites with more than 10 observations during the study period are selected."

Please describe a criterion for the site selection. In addition, the number of AERONET sites in Figure 10 seems much more than 10.

(16) 5. Discussions

First, the localization scale of 200km seems very small. The authors used a global model with a 223km horizontal resolution, which means that the authors simulated only synoptic scale phenomena and the distance of 200km was just a next grid. The data assimilation performs within only adjoining grids in this situation.

Second, the 400km localization is still too small to evaluate the localization length effect when the model resolution is 223km. If I were one of the authors, I would check the localization scales of 500km, 1000km, and 2000km. In addition, evaluation of the ensemble size might be needed when a large localization length is adopted.

Therefore, it is a hasty conclusion that the assimilation of aerosol vertical observation is "more" sensitive to the vertical localization than the horizontal localization.

(17) Lines 414-415

In this study, the hourly analyses were "validated" by only AERONET AOTs. CALIOP observations and MODIS AOTs merely confirmed the similarity between the assimilation results and the assimilated observations.

(18) Line 423

What does "all the four aerosol regimes" indicate here?

(19) Lines 428-430

If so, why did the authors not additionally conduct a simultaneous data assimilation of MODIS and CALIOP? It is unnatural that the simultaneous assimilation experiment is not at all mentioned in this manuscript.

Technical corrections:

(1) Line 320

"both the CALIOP or MODIS" -> "both the CALIOP and MODIS"

(2) Line 403

"MODIS observations, in" -> "MODIS observations. In"

(3) Caption of Figure 11

(g) CALIPSO derived vertical aerosol sub-types -> (i) CALIPSO derived vertical aerosol sub-types
* * *

---

## Author Comment (AC1) · 9 Sep 2019

The comment was uploaded in the form of a supplement:
https://www.atmos-chem-phys-discuss.net/acp-2019-497/acp-2019-497-AC1-supplement.pdf

---

## Author Comment (AC2) · 9 Sep 2019

**Response to the Comments of Referees**

**Investigating the assimilation of CALIPSO global aerosol vertical observations using Four-Dimensional Ensemble Kalman Filter**

Yueming Cheng, Tie Dai, Daisuke Goto, Nick A. J. Schutgens, Guangyu Shi, and Teruyuki Nakajima

We would like to thank to the two reviewers for giving constructive criticisms, which are very helpful in improving the quality of the manuscript. We have made major revision based on the critical comments and suggestions of the referees. The referee's comments are reproduced (black) along with our replies (blue) and changes made to the text (red) in the revised manuscript. All the authors have read the revised manuscript and agreed with submission in its revised form.

**Anonymous Referee #1**

**Comment NO.1:** *In this manuscript, the authors applied the 4D-LETKF and an aerosol model SPRINTARS online coupled with NICAM to generate hourly aerosol horizontal and vertical analyses for one-month using the CALIOP aerosol extinctions. The results are validated using both the CALIOP extinctions and the MODIS and AERONET AOTs observations, and the assimilation experiments are also evaluated in a statistical sense. Some interesting results have been found. It is probably the first study to conduct the hourly aerosol vertical extinctions assimilation using four-dimensional ensemble Kalman method for one month. The manuscript is well written and logically organized in its structure. I recommend to publish it after minor revision.*

**Response:** We thank the referee for this very positive assessment of our manuscript.

**Comment NO.2:** *In Section 2, all the data used in this paper should be described. For example, the CALIPSO level 2 VFM product in the 'quality control procedures', the CALIPSO level 3 monthly mean gridded aerosol profile products in Figure 9.*

**Response:** Accept.

**Changes in Manuscript:** We have added the descriptions of all the data in the revised manuscript.

1. The descriptions of CALPSO level 2 VFM product and the CALIPSO level 3 monthly

[revised manuscript text omitted]

**Changes in Manuscript:** We have rewritten the validation of the assimilation results to avoid the effects of the uncertainties of the CALIPSO and MODIS products on the conclusions in the revised manuscript. Please refer to the revised manuscript, Page 12-Page 16.

**Comment NO.4:** *Lines 349-357, Figure 9, The analysis should be repeated using CALIPSO Version 4 data, released last year, as the dust detection in Version 3 algorithms has been improved in Version 4. 1) In Version 3, any layer detected on single shots is classified as cloud. In Version 4, CAD is now applied to layers detected on single-shots. 2) The Version 3 CAD algorithm tends to classify elevated dust as cloud. This tendency is reduced in Version 4. The results 'the CALIOP AOTs are significantly lower over the western Saharan desert region' may be improved.*

**Response:** Agree. The dust detection in Version 3 has been improved in version 4. However, the CALIPSO level 3 aerosol profile products in Version 3 reported monthly mean profiles of aerosol optical properties are tropospheric products that reported averaged values at altitudes below 12 km, whereas the CALIPSO level 3 aerosol profile products in Version 4 include the stratospheric aerosol optical depths which calculated by integrating the mean extinction coefficient profile from the uppermost altitude bin (36 km) down to the altitude of the mean tropopause or to the last point above the tropopause of the extinction profile does not extend all the way to the tropopause. It means that the Version 3 aerosol products used in Figure 9 has not been updated to Version 4 until now, therefore we keep using the Version 3 aerosol products in the revised manuscript.

**Comment NO.5:** *Lines 14-15, Add 'the' before 'Cloud-Aerosol Lidar with Orthogonal*

*Polarization' and 'Cloud–Aerosol Lidar and Infrared Pathfinder Satellite Observation'.*

**Response:** Accept.

**Changes in Manuscript:** We have added 'the' before 'Cloud-Aerosol Lidar with Orthogonal Polarization' and 'Cloud–Aerosol Lidar and Infrared Pathfinder Satellite Observation'. Please refer to the revised manuscript, Page 1 Line 16-17.

**Comment NO.6:** *Line 43, Please give the full spelling when the abbreviation first appears. POLDER.*

**Response:** Accept.

**Changes in Manuscript:** We have given the full spelling 'POLarization and Directionality of the Earth's Reflectances' before 'POLDER'. Please refer to the revised manuscript, Page 2 Line 15-16.

**Comment NO.7:** *Line 44, Please give the full spelling when the abbreviation first appears, MODIS.*

**Response:** Accept.

**Changes in Manuscript:** We have given the full spelling 'Moderate Resolution Imaging Spectroradiometer' before 'MODIS'. Please refer to the revised manuscript, Page 2 Line 16-17.

**Comment NO.8:** *Line 55, Please give the full spelling when the abbreviation first appears, NIES.*

**Response:** Accept.

**Changes in Manuscript:** We have given the full spelling 'National Institute for Environment Studies' before 'NIES'. Please refer to the revised manuscript, Page 2 Line 31.

**Comment NO.9:** *Line 91, 'Moderate Resolution Imaging Spectroradiometer (MODIS)' should be changed to 'MODIS'.*

**Response:** Accept.

**Changes in Manuscript:** We have replaced 'Moderate Resolution Imaging Spectroradiometer (MODIS)' with 'MODIS'. Please refer to the revised manuscript, Page 4 Line 4.

**Comment NO.10:** *Lines 108-109, 'Takemura et al., 2000; Takemura et al., 2003; Takemura et al., 2009' -> 'Takemura et al., 2000, 2003, 2009' 'Satoh et al., 2005; Satoh et al., 2008; Satoh et al., 2014' -> 'Satoh et al., 2005, 2008, 2014'*

**Response:** Accept.

**Changes in Manuscript:** We have replaced 'Takemura et al., 2000; Takemura et al., 2003; Takemura et al., 2009' with 'Takemura et al., 2000, 2003, 2009' and replaced 'Satoh et al., 2005; Satoh et al., 2008; Satoh et al., 2014' with 'Satoh et al., 2005, 2008, 2014'. Please refer to the revised manuscript, Page 4 Line 34-35.

**Comment NO.11:** *Line 140, 'the horizontal and the vertical localization factors' should be changed to 'the horizontal and vertical localization factors'.*

**Response:** Accept.

**Changes in Manuscript:** We have replaced 'the horizontal and the vertical localization factors' with 'the horizontal and vertical localization factors'. Please refer to the revised manuscript, Page 6 Line 36.

**Comment NO.12:** *Line 147, 'over November 2016' should be changed to 'in November 2016'.*

**Response:** Accept.

**Changes in Manuscript:** We have replaced 'over November 2016' with 'in November 2016'. Please refer to the revised manuscript, Page 7 Line 13.

**Comment NO.13:** *Line 337, the grammatical error exists in 'the both the two assimilations correctly…'*

**Response:** Accept.

**Changes in Manuscript:** We have delete this wrong sentence.

**Comment NO.14:** *Line 365, 'the South and East Asia' should be changed to 'South and East Asia'*

**Response:** Accept.

**Changes in Manuscript:** We have replaced 'the South and East Asia' with 'South and East Asia'. Please refer to the revised manuscript, Page 13 Line 2.

**Comment NO.15:** *Line 368, 'both the CALIOP and the MODIS assimilations' should be changed to 'both the CALIOP and MODIS assimilations'*

**Response:** Accept.

**Changes in Manuscript:** We have replaced 'both the CALIOP and the MODIS assimilations' with 'both the CALIOP and MODIS assimilations'. Please refer to the revised manuscript, Page 13 Line 14.

**Anonymous Referee #2**

**Comment NO.1:** *The authors present the result of an aerosol analysis comparison between CALIOP/CALIPSO data assimilation and NRL-MODIS data assimilation using a 4-dimensional ensemble Kalman filter with a global aerosol model to demonstrate the effects of the vertical profile information on the analysis performance. The manuscript looks well written and organized in its structure. Some interesting results are included for data assimilation researchers. However, it is impugnable that the "hourly" aerosol vertical analysis is really scientifically plausible, although the authors claim the hourly analysis as the novelty of this paper.*

**Response:** Thank you for your suggestions. In the original manuscript, we didn't clearly describe our 4D-LETKF assimilation method. In the revised manuscript, we have added the detailed descriptions of our implement method to prove the scientificity of the hourly aerosol vertical analysis.

[revised manuscript text omitted]

**Changes in Manuscript:** As for the descriptions of the assimilation methodology, please refer to the revised manuscript, Page 5 Line 28-Page 6 Line 28.

**Comment NO.2:** *In addition, some of the validation processes the authors used were not suitable to validate data assimilation results.*

**Response:** Thank you for your suggestions. In order to validate the data assimilation results by the suitable validation processes, we have reconstructed the experiments and the validation processes.

As summarized in Table 1, a total of five numerical experiments are conducted for this study. A single deterministic simulation with the default model configuration is performed as a reference in November 2016 (a free running, FR experiment hereafter). The two assimilation experiments assimilating the CALIOP vertical extinction coefficients with time windows of 1 and 24 hr (hereafter called the LETKF-CALIPSO and 4D-LETKF-24H-CALIPSO, respectively) are performed to investigate the influences of the assimilation time window on the hourly aerosol analysis. Combined with the 4D-LETKF-24H-CALIPSO experiment, two more assimilation experiments are designed to investigate the impacts of assimilating the observations whether including the vertical information and the impacts of the multi-sensor data assimilation on the model simulations. The first one assimilates the NRL MODIS AOTs including no vertical aerosol information (4D-LETKF-24H-NRL experiment hereafter). In the second one, the CALIOP vertical extinction coefficients and NRL MODIS AOTs are assimilated simultaneously.

Table 1: Experimental design for the sensitivity tests in this study

| Sensitivity experiments |
| --- |
| FR: single deterministic forecast without aerosol data assimilation |
| LETKF-CALIPSO: assimilates the CALIPSO vertical extinction coefficients and the cycle frequency and forecast length are set to 1 hr |

4D-LETKF-24H-CALIPSO: same as LETKF-CALIPSO except the cycle frequency and forecast length are set to 24 hr with hourly model output and analysis

4D-LETKF-24H-NRL: same as 4D-LETKF-24H-CALIPSO except assimilates the NRL MODIS level-3 gridded AOTs

4D-LETKF-24H-CALIPSO-NRL: same as 4D-LETKF-24H-CALIPSO except assimilates both the CALIPSO vertical extinction coefficients and NRL MODIS level 3 gridded AOTs

The results in the FR, LETKF-CALIPSO, and 4D-LETKF-24H-CALIPSO experiments are compared with the assimilated CALIOP vertical extinctions as the self-verification. The results in the 4D-LETKF-NRL experiment are compared with the assimilated NRL MODIS AOTs as the self-verification. In order to investigate the differences of AOT simulations among the four assimilation experiments, we use the independent AERONET AOT observations and the independent MODIS Aqua AOTs through screening out the portions of the MODIS Aqua AOT products which are same as the assimilated NRL MODIS dataset. To independently validate the aerosol vertical extinctions, we assimilated the night-time CALIOP aerosol extinctions and then use the remaining CALIOP observations in the day-time for validation.

**Changes in Manuscript:** As for the descriptions of the new experimental design, please refer to the revised manuscript, Page 7 Line 34-Page 8 Line 8. As for the new validation processes, please refer to Section 4 in the revised manuscript.

**Comment NO.3:** *The authors insist that this is the first study to conduct the hourly aerosol analysis using a four-dimensional data assimilation method. That is because an hourly data assimilation with a 24-hour assimilation window is mathematically inadequate, and thus, nobody has conducted it.*

*If I am not mistaken, the authors assimilate one observation again and again (probably 25 times) because the data assimilation is performed every hour and each time window is set 24 hours. In contrast, if the authors define the 1, 2,… 23-hour forecasts as the analysis, which means the data assimilation is done once per day, the authors' method is completely the same as the previous studies.*

*Theoretically, observations should be used strictly only once in the data assimilation process. Of course, in reality, a multiple use of one observation sometimes yields a good analysis result. However, those practical methods are often controversial. At least, it is not scientifically*

*ingenious. The authors did not discuss this controversial issue at all in the manuscript. For example, Eugenia Kalnay's "Running in Place (RIP)" scheme (https://doi.org/10.1002/qj.652), which is slightly similar to the authors' method, assimilates one observation repeatedly. Therefore, Kalnay et al. carefully discussed the RIP scheme and its application range in their papers. Despite that, the RIP scheme has been controversial.*

*If the authors like to generate hourly aerosol analyses with hourly observations, a three dimensional ensemble Kalman filter should be used to avoid a multiple use of one observation, or very narrow time localization should be applied to a four-dimensional ensemble Kalman filter to reduce the influence of temporally remote observations. In that case, a 24-hour assimilation window is too long.*

**Response:** As shown in Fig.1 in the revised manuscript, in our implement the assimilation system performs the ensemble forecast for 24 hours and outputs at every hour time slot within the time window. Based on the innovations throughout the assimilation window, the ensemble mean background observations and the background ensemble perturbation matrix are formed at the various time slots when the observations are available and then vertically concatenated to form a combined background observation mean $\bar{y}^b$ and perturbation matrix $Y^b$. Following to the LETKF formulas, all the innovations $(y^o - \bar{y}^b)$ and $Y^b$ throughout the day are used for the calculation of the weight matrix $\bar{w}^a$ and $W^a$. The weights determined at the end of a short assimilation window (e.g., 24 hours) should be valid throughout the window. To perform a linear combination of ensemble trajectories, the same $\bar{w}^a$ is then applied to the state vector at every hour time slot throughout the assimilation window to obtain the hourly analysis ensemble mean. The analyzed aerosol fields at the last slot can then directly serve as the initial conditions for the next 24 hours of forward simulation, therefore, our implement of the 4D-LETKF can also avoid a multiple use of one observation without overlapping the ensemble forecasts between adjacent assimilation cycles.

We also conduct an experiment assimilating the CALIPSO vertical extinction coefficients using the LETKF (called LETKF-CALIPSO) that generally same as the previous studies. We have carefully compared the results assimilating the CALIOP aerosol extinction observations using the 4D-LETKF and the LETKF. Through the internal checks, the performances of the 4D-LETKF experiment are generally comparable to those of the LETKF experiment. This indicates

the weights determined at the end of the 24 hours assimilation window are valid to optimize the ensemble trajectories and the temporally remote asynchronous observations within 24 hours have limited influence on the analysis. Moreover, the elapsed time for the one-month assimilation over November 2016 with 4D-LETKF-24H-CALIPSO is much shorter than that of the LETKF-CALIPSO experiment. This is due to the 4D-LETKF can avoid frequent switching between the assimilation and model ensemble forecasts. Through assimilating the night-time CALIOP aerosol vertical extinctions and using the remaining CALIOP observations in the day-time for independent validation, we have found that the 4D-LETKF experiment is significantly superior to the LETKF experiment. Those results demonstrate that the 4D-LETKF implemented in this study is mathematically adequate.

**Comment NO.4:** *Although the MODIS-Aqua DT/DB dataset is described in Section 2.2, the NRL MODIS dataset is not. Although it is described in Section 3.2 how to make super observations of CALIOP/CALIPSO, those of NRL MODIS are not described. Please clarify them. The NRL MODIS dataset is probably 6-hourly. If so, how did you make the hourly analysis of DA-NRL?*

**Response:** Accept. We have added the descriptions of the NRL MODIS dataset in the revision manuscript. In this study, the United States Naval Research Laboratory (NRL) quality-assured and controlled MODIS level 3 AOT products are also used for data assimilation. The NRL MODIS AOTs are based on the MODIS level 2 aerosol products and aggregated to 1°×1° grid with reduced biases and error variances for using in the near-real-time data assimilation processes. The datasets consist of 6-hourly gridded AOTs and error estimates at four times per day (00:00, 06:00, 12:00, and 18:00 UTC).

Due to the representation of the observations are considered during the development of the NRL MODIS datasets and the NRL MODIS AOTs have been subjected to extensive quality assurance and quality check procedures for aerosol assimilation, we directly use the NRL MODIS AOTs and the corresponding AOT uncertainties every 6-hour at 1°.

In this study, we use a 24 h assimilation window and the NRL MODIS AOT observations available at 00:00, 06:00, 12:00, and 18:00 UTC are considered for data assimilation within the window as same as Di Tomaso et al. (2017). The LETKF implementation with a four dimensional extension as described in Hunt et al. (2007) allows observations gathered at

different times to be assimilated simultaneously in a natural way. The assimilation system uses a 1-day forecast as first guess with hourly output. Based on the innovations throughout the assimilation window, the ensemble mean background observations and the background ensemble perturbation matrix are formed at the various time slots when the observations are available and then vertically concatenated to form a combined background observation mean $\bar{y}^b$ and perturbation matrix $Y^b$. $\bar{y}^b$ and $Y^b$ are used for the calculation of the weight matrix $\overline{w}^a$ based on all the innovations throughout the day. This $\overline{w}^a$ is then applied to the state vector at every hour time slot throughout the assimilation window to obtain the hourly analysis ensemble mean of 4D-LETKF-24H-NRL experiment.

**Comment NO.5:** *Because the 4D-LETKF is not a smoother, the analysis has only one timeslot during its time window, which can be arbitrarily specified. When is the analysis timeslot during the 24-hour window in this study?*

*The control variables are fine/coarse mass mixing ratios only? The authors did not separately control all the prognostic variables of NICAM-SPRINTARS? Please describe it in detail.*

**Response:** As shown in Fig.1 in the revised manuscript, The analysis target time is chosen to be at every hour time slot within the 24-hour window. In our implement, we performs the ensemble forecast for 24 hours and outputs at every hour time slot within the time window. Based on the innovations throughout the assimilation window, the ensemble mean background observations and the background ensemble perturbation matrix are formed at the various time slots when the observations are available and then vertically concatenated to form a combined

background observation mean $\bar{y}^b$ and perturbation matrix $Y^b$. Following to the LETKF formulas, all the innovations $(y^o - \bar{y}^b)$ and $Y^b$ throughout the day are used for the calculation of the weight matrix $\bar{w}^a$ and $W^a$. The weights determined at the end of a short assimilation window (e.g., 24 hours) should be valid throughout the window. To perform a linear combination of ensemble trajectories, the same $\bar{w}^a$ is then applied to the state vector at every hour time slot throughout the assimilation window to obtain the hourly analysis ensemble mean. The analyzed aerosol fields at the last slot can then directly serve as the initial conditions for the next 24 hours of forward simulation, therefore, our implement of the 4D-LETKF can also avoid a multiple use of one observation without overlapping the ensemble forecasts between adjacent assimilation cycles.

The control variables are fine and coarse mass mixing ratios and we did not separately control all the prognostic variables of NICAM-SPRINTARS to better reduce the computational resources and limit the model information to a few modes only. The subspecies of the SPRINTARS predicted aerosols are summarized into a fine (carbonaceous and sulfate aerosol) and a coarse (sea salt and dust aerosol) mode for assimilation. The modeled aerosol fine and coarse mass mixing ratios are hourly optimized using the hourly aggregated observations during the assimilation window and the mass mixing ratios after data assimilation for the each subspecies are determined from their relative fractions before assimilation.

**Changes in Manuscript:** Please refer to the revised manuscript, Page 7 Line 6-12.

**Comment NO.6:** *The horizontal and vertical localization scales are described here, but the temporal localization scale is not. If the temporal localization is not applied, please specify it.*

**Response:** The temporal localization is not applied and there are two reasons account for the negligence of the temporal localization in this study.

1) Firstly, the lifetimes of the various aerosols are generally more than 1-day which longer than the 24-hour assimilation window we used.

2) Second, we have conducted a new experiment using LETKF (named LETKF-CALIPSO), and it is basically equivalent to the 4D-LETKF-24H-CALIPSO experiment with 1-hour time localization. We have carefully compared the results assimilating the CALIOP aerosol extinction observations using the 4D-LETKF and the LETKF. Through the internal checks, the performances of the 4D-LETKF-24H-CALIPSO experiment are generally comparable

to those of the LETKF-CALIPSO experiment. This indicates the weights determined at the end of the 24 hours assimilation window are valid to optimize the ensemble trajectories and the temporally remote asynchronous observations within 24 hours have limited influence on the analysis. Through assimilating the night-time CALIOP aerosol vertical extinctions and using the remaining CALIOP observations in the day-time for independent validation, we have found that the 4D-LETKF experiment is significantly superior to the LETKF experiment. Therefore, we have not applied the temporal localization in this study.

**Changes in Manuscript:** Please refer to the revised manuscript, Page 7 Line 4-6.

**Comment NO.7:** *The authors say "… the DA-CALIPSO experiment achieves more higher correlations and lower RMS differences than those with the MODIS assimilation. This indicates that …, while the CALIOP assimilation can also improve the temporal and spatial variations of the aerosol extinctions," but I do not agree. The authors validated the two experiments with the CALIOP data that were assimilated only for the DA-CALIPSO experiment. The reason of the high correlation and lower RMSD of the DA-CALIPSO is the self-verification between the DA-CALIPSO results and the CALIOP data, which should be strongly refrained for data assimilation validation. What the authors did in Section 4.1 was sanity or internal checks, not validation.*

**Response:** Agree. We have deleted these wrong sentences here and compare the extinction analysis in the LETKF-CALIPSO and 4D-LETKF-24H-CALIPSO experiments to the assimilated CALIOP observations as the internal checks in Section 4.1. The results of internal checks show that the assimilation system is operated successfully. To independently validate the aerosol vertical extinctions, we assimilated the night-time CALIOP aerosol extinctions and then use the remaining CALIOP observations in the day-time for validation.

**Changes in Manuscript:** Please refer to Section 4.1 and 4.4 in the revised manuscript.

**Comment NO.8:** *The DA-CALIPSO is not "clearly" superior to the FR and DA-NRL because in this section the authors merely checked the sanity of the DA-CALIPSO experiment. If the authors like to indicate the superiority of the DA-CALIPSO or DA-NRL results, independent observations should be used. In other way, the forecasts from the data assimilation analyses can be validated by the assimilated observations.*

**Response:** Agree. We have deleted these wrong sentences in this section. In order to investigate

the differences of AOT simulations among the four assimilation experiments, we use the independent AERONET AOT observations and the independent MODIS Aqua AOTs through screening out the portions of the MODIS Aqua AOT products which are same as the assimilated NRL MODIS dataset. To independently validate the aerosol vertical extinctions, we assimilated the night-time CALIOP aerosol extinctions and then use the remaining CALIOP observations in the day-time for validation. It is found that both the CALIOP and MODIS assimilation can improve the magnitude of the simulated aerosol extinctions, however the CALIOP assimilation is superior to the MODIS assimilation in terms of modifying the incorrect aerosol vertical distributions and reproducing the real magnitudes and variations. Compared with the independent CALIOP extinction observations in the day-time over the ocean, the 4D-LETKF CALIOP assimilation is better than that of the MODIS assimilation. It indicates the optimized aerosol vertical distributions are more benefited from the CALIOP vertical observations than the MODIS column intergraded observations. The simultaneous CALIOP and MODIS assimilation experiment has the best performance. This is probably due to the aerosol vertical distributions, which are unable to be optimized by assimilating the sparse CALIOP observations, are further optimized by the MODIS observations.

**Changes in Manuscript:** Please refer to Section 4.2, 4.3, and 4.4 in the revised manuscript.

**Comment NO.9:** *Does this mean that the vertical data assimilation breaks the aerosol profile balance? The analysis of the aerosol profile becomes not similar to either the prior or observation information? This is very interesting.*

**Response:** Agree. We apply the vertical localization in our assimilation system to avoid the effect of noise in the ensemble statistics due to its limited size. The sets of the observations used for a pair of neighboring grid points overlap heavily, the local weights $w^a$ for the two grid points are similar. Therefore, the neighboring grid points do not yield very different analysis. If the grid beyond the truncated localization tails, the observations can not affect it. This may slightly break the aerosol profile balance. If the model requires higher order smoothness, it may be necessary to post-process the analysis ensemble members to smooth them. In this study, we didn't apply the post processes to smooth the aerosol vertical profile.

**Comment NO.10:** *The word "improvement" is not appropriate. The authors were simply tested the similarity between the DA-CALIPSO results and the assimilated observations in this*

*section.*

**Response:** Accept. The comparison between the results of experiments with CALIOP assimilation and the assimilated observations is the internal check, therefore we are unable to conclude which regime has the most efficient improvement among the four regimes.

**Changes in Manuscript:** We have replaced the "most efficient improvements" with "highest assimilation efficiencies", please refer to the revised manuscript, Page 11 Line 10.

**Comment NO.11:** *As the authors say, the MODIS and CALIOP observations are generally inconsistent. Therefore, the authors have to compare the MODIS and CALIOP observations used in this study before the validation of data assimilation results.*

**Response:** According to previous studies such as Ma et al. (2013), the spatial distribution and seasonal variability of CALIPSO AOD is generally consistent with that of MODIS. Therefore, we apply the MODIS Aqua AOTs to validate the assimilation results. We only found the significant discrepancies in the western Saharan Desert, thus we compare the multi-annual averaged aerosol optical thicknesses (AOTs) for the MODIS and CALIOP AOTs dataset. The detailed comparisons of AOTs between CALIPSO and MODIS are out of the scope in this study.

References:

Ma, X., Bartlett, K., Harmon, K. and Yu, F.: Comparison of AOD between CALIPSO and MODIS: significant differences over major dust and biomass burning regions, Atmospheric Measurement Techniques, 6(9), 2391–2401, doi:10.5194/amt-6-2391-2013, 2013.

**Comment NO.12:** *Nobody can conclude that the CALIOP assimilation is "superior" to the MODIS assimilation before the validation with independent observations. In this section, the authors checked only the similarity between the CALIOP assimilation results and the assimilated CALIOP observations.*

**Response:** Agree. We have deleted this wrong statement. To independently validate the aerosol vertical extinctions, we assimilated the night-time CALIOP aerosol extinctions and then use the remaining CALIOP observations in the day-time for validation. Compared with the independent CALIOP extinction observations in the day-time over the ocean, the 4D-LETKF CALIOP assimilation is better than that of the MODIS assimilation. It indicates the optimized aerosol vertical distributions are more benefited from the CALIOP vertical observations than

the MODIS column intergraded observations.

**Changes in Manuscript:** Please refer to Section 4.4 in the revised manuscript.

**Comment NO.13:** *Section 4.1 is too long in comparison to Sections 4.2 and 4.3. The most important result is Section 4.3. Although Section 4.1 is much longer than Sections 4.2 and 4.3, it does not contain the similarity check between the data assimilation results and the NRL-MODIS data.*

**Response:** Agree. The results in the FR, LETKF-CALIPSO, and 4D-LETKF-24H-CALIPSO experiments are compared with the assimilated CALIOP vertical extinctions as the self-verification in the Section 4.1.1. The results in the 4D-LETKF-NRL experiment are compared with the assimilated NRL MODIS AOTs as the self-verification in the Section 4.1.2. In order to investigate the differences of AOT simulations among the four assimilation experiments, we use the independent AERONET AOT observations (Section 4.2) and the independent MODIS Aqua AOTs (Section 4.3) through screening out the portions of the MODIS Aqua AOT products which are same as the assimilated NRL MODIS dataset. To independently validate the aerosol vertical extinctions, we assimilated the night-time CALIOP aerosol extinctions and then use the remaining CALIOP observations in the day-time for validation (Section 4.4).

**Changes in Manuscript:** Please refer to Section 4.1, 4.2, and 4.3 in the revised manuscript.

**Comment NO.14:** *The authors described a comparison with the MODIS Aqua C6.1 AOT products as an independent validation. Probably it is false. The NRL MODIS dataset highly probably contains information very close to the MODIS Aqua C6.1 AOT products. If the authors like to insist that it is an independent validation, the portion originated from MODIS Aqua should be screened out before the NRL MODIS data assimilation.*

**Response:** Agree. In Section 4.3, in order to describe a comparison with the MODIS Aqua AOT products as an independent validation, we screen out the portions of the MODIS Aqua AOT products which are same as the NRL MODIS dataset to avoid the overlap information.

**Changes in Manuscript:** Please refer to Section 4.3 in the revised manuscript.

**Comment NO.15:** *I do not understand the necessity to eliminate the potential effects of the sparse CALIPSO observations. I think the authors would rather evaluate the effect of the CALIOP sparseness here.*

**Response:** Due to the sparse CALIOP observations and the localization used in the assimilation

system, the simulated AOTs in many model grids are unable to be affected by the aerosol data assimilation. Therefore, we only analyze the simulated AOTs with more than 30% variations between the 4D-LETKF-24H-CALIPSO and FR experiments. We include the evaluation of the effect of the CALIOP sparseness in Section 5.

**Changes in Manuscript:** Please refer to the revised manuscript, Page 12 Line 35-38.

**Comment NO.16:** *It is not unexpectable that DA-CALIPSO has larger biases and RMSEs compared to FR. Generally, different remote instruments have large biases. Therefore, "deterioration" is a hasty conclusion. A careful evaluation of the MODIS AOT and CALIOP AOT must be done with independent observations other than Figure 9 to say "deterioration".*

**Response:** Agree.

**Changes in Manuscript:** We have replaced the "unexpected" with "found" and replaced the "such deterioration is not found in the DA-NRL experiment" with "such discrepancy is not found between the FR and 4D-LETKF-24H-NRL experiments", please refer to the revised manuscript, Page 14 Line 16 and Page 14 Line 18-19.

**Comment NO.17:** *Line 362 "The AERONET sites with more than 10 observations during the study period are selected." Please describe a criterion for the site selection. In addition, the number of AERONET sites in Figure 10 seems much more than 10.*

Response: We select an AERONET site if it has simultaneously at least 10 hours in one month (not necessarily consecutive) where the hourly AOTs is not missing. We require at least 10 observations in order to avoid selecting sites where the statistics are statistically non-significant. A total of 191 AERONET sites are selected for comparison.

**Changes in Manuscript:** Please refer to the revised manuscript, Page 12 Line 31-35.

**Comment NO.18:** *First, the localization scale of 200km seems very small. The authors used a global model with a 223km horizontal resolution, which means that the authors simulated only synoptic scale phenomena and the distance of 200km was just a next grid. The data assimilation performs within only adjoining grids in this situation.*

*Second, the 400km localization is still too small to evaluate the localization length effect when the model resolution is 223km. If I were one of the authors, I would check the localization scales of 500km, 1000km, and 2000km. In addition, evaluation of the ensemble size might be needed when a large localization length is adopted.*

*Therefore, it is a hasty conclusion that the assimilation of aerosol vertical observation is "more" sensitive to the vertical localization than the horizontal localization.*

**Response:** In this study, the horizontal localization factor is defined following Gaussian shapes as $exp(-r^2/2\sigma^2)$, where the $\sigma$ represents the localization length and the $r$ is the distance of observations from the local patch center. Although the Gaussian function has infinitely long tails, we truncate the tails to simulate the fifth-order piecewise rational function, which is widely used localization weighting function in the EnKF studies. The fifth-order rational function drops to zero at $r = 2 \cdot \sqrt{10/3} \cdot \sigma$ and we do not assimilation observations beyond this distance. Therefore, we assimilate the observations within 730 km with the localization scale of 200 km, this corresponds to about three adjoining grids as shown in Fig. A1.

[Figure]

Figure A1. A schematic showing the horizontal observation localization. The observations located within the red circle as the blue diamonds are used for assimilation. The observations located out of the red circle as the green diamonds are not used for assimilation.

We have conducted three more assimilation experiments named 4D-LETKF-24H-CALIPSO-HL500, 4D-LETKF-24H-CALIPSO-HL1000, and 4D-LETKF-24H-CALIPSO-HL2000 experiments. The experiments are in the same model configuration as that of the 4D-LETKF-24H-CALIPSO experiment except the horizontal localization lengths of 500 km, 1000km, and 2000km used in the 4D-LETKF-24H-CALIPSO-HL500, 4D-LETKF-24H-CALIPSO-HL1000,

4D-LETKF-24H-CALIPSO-HL2000 experiments. As shown in Fig. 16 in the revised manuscript, it is found that the 4D-LETKF-24H-CALIPSO experiment shows the highest percentages of deviations between ±0.05 and ±0.10. By the increments of the horizontal localization length, the percentages of deviations between ±0.05 and ±0.10 are decreasing and the peaks of the PDF plots tend to far away from 0. Therefore, we use the 200 km horizontal localization length (i.e., observations located within 730 km are assimilated) for the aerosol data assimilation in this study.

Agree. We have deleted the statement "the assimilation of aerosol vertical observation is more sensitive to the vertical localization than the horizontal localization" .

**Comment NO.19:** *In this study, the hourly analyses were "validated" by only AERONET AOTs. CALIOP observations and MODIS AOTs merely confirmed the similarity between the assimilation results and the assimilated observations.*

**Response:** Done. The results in the FR, LETKF-CALIPSO, and 4D-LETKF-24H-CALIPSO experiments are compared with the assimilated CALIOP vertical extinctions as the self-verification in the Section 4.1.1. The results in the 4D-LETKF-NRL experiment are compared with the assimilated NRL MODIS AOTs as the self-verification in the Section 4.1.2. In order to investigate the differences of AOT simulations among the four assimilation experiments, we use the independent AERONET AOT observations (Section 4.2) and the independent MODIS Aqua AOTs (Section 4.3) through screening out the portions of the MODIS Aqua AOT products which are same as the assimilated NRL MODIS dataset. To independently validate the aerosol vertical extinctions, we assimilated the night-time CALIOP aerosol extinctions and then use the remaining CALIOP observations in the day-time for validation (Section 4.4).

**Changes in Manuscript:** Please refer to Section 4 in the revised manuscript.

**Comment NO.20:** *What does "all the four aerosol regimes" indicate here?*

**Response:** The "four aerosol regimes" represents the biomass burning, dust, industrial, and maritime downwind regimes defined in the caption of Fig. 3 in the revised manuscript.

**Comment NO.21:** *If so, why did the authors not additionally conduct a simultaneous data assimilation of MODIS and CALIOP? It is unnatural that the simultaneous assimilation experiment is not at all mentioned in this manuscript.*

**Response:** Accepted and Done. We have also conducted an assimilation experiment

simultaneously assimilating the observations from the MODIS and CALIOP (named 4D-LETKF-24H-CALIPSO-NRL) in the revised manuscript.

**Comment NO.22:** *"both the CALIOP or MODIS" -> "both the CALIOP and MODIS"*

**Response:** Done.

**Changes in Manuscript:** We have replaced "both the CALIOP or MODIS" with "both the CALIOP and MODIS". Please refer to the revised manuscript, Page 16 Line 1.

**Comment NO.23:** *"MODIS observations, in" -> "MODIS observations. In"*

**Response:** Done.

**Changes in Manuscript:** We have replaced "MODIS observations, in" with "MODIS observations. In". Please refer to the revised manuscript, Page 16 Line 26.

**Comment NO.24:** *(g) CALIPSO derived vertical aerosol sub-types -> (i) CALIPSO derived vertical aerosol sub-types*

**Response:** Done.

**Changes in Manuscript:** We have replaced "(g) CALIPSO derived vertical aerosol sub-types" with "(m) CALIPSO derived vertical aerosol sub-types". Please refer to the Fig. 10 in the revised manuscript.

[revised manuscript text omitted]

---

## Author Response (AR2)

**Response to the Comments of Referees**

**Investigating the assimilation of CALIPSO global aerosol vertical observations using Four-Dimensional Ensemble Kalman Filter**

Yueming Cheng, Tie Dai, Daisuke Goto, Nick A. J. Schutgens, Guangyu Shi, and Teruyuki Nakajima

We would like to thank to the two reviewers for giving constructive criticisms again concerning our manuscript. Based on the critical comments and suggestions, we have made careful modification on the manuscript. The referee's comments are reproduced (**black**) along with our replies (**blue**) and changes made to the text (**red**) in the revised manuscript. All the authors have read the revised manuscript and agreed with submission in its revised form.

**Anonymous Referee #1**

**Comment NO.1:** The author has made significant improvements to the manuscripts. I recommend accepting it after modifying several technical corrections as indicated below: Please pay attention to the grammar in the sentences of the manuscript.

**Response:** We are extremely grateful to the referee for the positive assessment of our manuscript.

**Comment NO.2:** Page 4 Line 10, 'Due to' should be changed to 'Because' or 'Since'. Similar situations often occur in the manuscript (Page 7 Line 5; Page 15 Line 3, 29, 37; Page 17 Line 28).

**Response:** Accept.

**Changes in Manuscript:** We have replaced the 'Due to' with 'Because' and 'Since'. Please refer to the revised manuscript, Page 4 Line 11, Page 7 Line 6, Page 15 Line 7, Page 15 Line 33, Page 16 Line 1, and Page 17 Line 29.

**Comment NO.3:** Page 12 Line 33, 'where' should be changed to 'and'.

**Response:** Agree. To make the statement clearer, we have replaced this sentence with 'In order to make the statistics significant, we require at least 10 observations in each selected site'.

**Changes in Manuscript:** Please refer to the revised manuscript, Page 13 Line 1-2.

**Anonymous Referee #2**

**Comment NO.1:** Satellite derived data is often considered as one of the most important data sources for aerosol DA due to its wide coverage. Many DA algorithms have been developed to assimilate MODIS AOD data to obtain better initial fields for aerosol short forecast and generate the aerosol analyses for research community. However, the MODIS AOD data is just the integration of extinction coefficient that is difficult to improve the vertical profile of aerosol. In this paper, the authors assimilated the CALIPSO vertical extinction coefficients observations to improve the simulations of the aerosol vertical distributions using a global chemical transport model. The assimilation methodology was based on the four-dimensional Local Ensemble Transform Kalman Filter (4D-LETKF), and several experiments were conducted to investigate the impact of the assimilation system parameters and observation data on the assimilation results. The work presented in this study is innovative and significant. The paper is in general well written and the structure is clear and logical. The authors also provided clear explanations of why the results seen may be occurring. I recommend accepting for publication following minor revision.

**Response:** We thank the referee for the valuable feedback of our manuscript.

**Comment NO.2:** Page 6 Line 27, 'implement' -> 'implementation'?

**Response:** Accept.

**Changes in Manuscript:** We have replaced the 'implement' with 'implementation'. Please refer to the revised manuscript, Page 6 Line 28.

**Comment NO.3:** Page 7 Line 8, Please clarify the observation operator to map the aerosol mixing ratio to the optical properties.

**Response:** Done. The observation operators we used to map the model state vector into the aerosol extinction coefficient $\sigma$ and the AOT observation space at wavelength $\lambda$ are calculated as $\sigma_j(\lambda) = \beta(\lambda) \cdot M_j$ and $AOT(\lambda) = \sum_{j=1}^{n} (\beta(\lambda) \cdot M_j)$, where the $M_j$ represents the simulated aerosol dry mass concentration in each model level $j$, $\{j = 1,2,\ldots,n\}$ and the $\beta$ represents the prescribed aerosol mass extinction coefficient.

**Changes in Manuscript:** We have clarified the descriptions of the observation operator in the revised manuscript, please refer to the revised manuscript, Page 7 Line 13-18.

**Comment NO.4:** The assimilation cycle used in Figure 1 is a bit confusing.

**Response:** Accept.

**Changes in Manuscript:** We have replaced the confused 'assimilation cycle' with 'Assimilation Cycle 1 (2016-11-01)', 'Assimilation Cycle 2 (2016-11-02)', and 'Assimilation Cycle 3 (2016-11-03)'. Please refer to the Fig.1 in the revised manuscript.

**Comment NO.5:** Page 8 Line 3, 'are' -> 'is'

**Response:** Accept.

**Changes in Manuscript:** We have replaced the 'are' with 'is'. Please refer to the revised manuscript, Page 8 Line 1.

**Comment NO.6:** Page 10 Line 2, delete 'the' before '0.30'.

**Response:** Accept.

**Changes in Manuscript:** We have deleted the 'the' before '0.30'. Please refer to the revised manuscript, Page 10 Line 8.

**Comment NO.7:** Page 10 Line 5 and Line 29, 'relative larger' -> 'relatively larger'.

**Response:** Accept.

**Changes in Manuscript:** We have replaced the 'relative' with 'relatively'. Please refer to the revised manuscript, Page 10 Line 11 and Page 10 Line 35.

**Comment NO.8:** Page 13 Line 20, 'which in also mentioned in Zhang et al. (2014)' -> 'which is also mentioned in Zhang et al. (2014)'.

**Response:** Accept.

**Changes in Manuscript:** We have replaced the 'which in also mentioned in Zhang et al. (2014)' with 'which is also mentioned in Zhang et al. (2014)'. Please refer to the revised manuscript, Page 13 Line 25.

**Comment NO.9:** Page 15 Line 32, 'With respected to' -> 'With respect to'.

**Response:** Accept.

**Changes in Manuscript:** We have replaced the 'With respected to' with 'With respect to'. Please refer to the revised manuscript, Page 15 Line 36.